# A rare case of brominated small molecule acceptors for high-efficiency organic solar cells

Huazhe Liang[1,5], Xingqi Bi[1,5], Hongbin Chen[1], Tengfei He[1], Yi Lin[2], Yunxin Zhang[3], Kangqiao Ma[1], Wanying Feng[1], Zaifei Ma [2], Guankui Long [3], Chenxi Li[1], Bin Kan[3], Hongtao Zhang[1], Oleg A. Rakitin [4], Xiangjian Wan[1], Zhaoyang Yao [1] ✉ & Yongsheng Chen [1] ✉

Given that bromine possesses similar properties but extra merits of easily synthesizing and polarizing comparing to homomorphic fluorine and chlorine, it is quite surprising very rare high-performance brominated small molecule acceptors have been reported. This may be caused by undesirable film morphologies stemming from relatively larger steric hindrance and excessive crystallinity of bromides. To maximize the advantages of bromides while circumventing weaknesses, three acceptors (CH20, CH21 and CH22) are constructed with stepwise brominating on central units rather than conventional end groups, thus enhancing intermolecular packing, crystallinity and dielectric constant of them without damaging the favorable intermolecular packing through end groups. Consequently, PM6:CH22-based binary organic solar cells render the highest efficiency of 19.06% for brominated acceptors, more excitingly, a record-breaking efficiency of 15.70% when further thickening active layers to ~500 nm. By exhibiting such a rare high-performance brominated acceptor, our work highlights the great potential for achieving record-breaking organic solar cells through delicately brominating.

In view of the deeper understanding of molecule design, device engineering and charge transfer/transport mechanism[1–9], organic solar cells (OSCs) have undergone a blowout growth in the past few decades, leading to power conversion efficiencies (PCEs) of single-junction OSCs surpassing 19%[10–15] and tandem OSCs over 20%[16,17], respectively. At present, the continuous exploration of small molecule acceptors (SMAs) with a distinctive "acceptor-donor-acceptor" (A-D-A) architecture is still the most concerned issue if more efficient OSCs are further expected[3,18]. Note that the desirable three-dimensional (3D) intermolecular packing network, which will be in favor of superior charge generation/transport/recombination dynamics, is highly desired for well-established SMAs[19–21]. That is why further optimizing the molecular packing of SMAs through delicate structural tailoring is crucially important for achieving record-breaking OSCs[3,22]. Among the various strategies to tune molecule packing[3,21,23–25], halogenation on end units of SMAs has been the most employed but also indispensable one, especially in the state-of-the-art OSCs[26–28]. As it has been proven that halogenation plays a crucial role in (1) greatly tuning molecular energy

[1]State Key Laboratory and Institute of Elemento-Organic Chemistry, The Centre of Nanoscale Science and Technology and Key Laboratory of Functional Polymer Materials, Renewable Energy Conversion and Storage Center (RECAST), College of Chemistry, Nankai University, 300071 Tianjin, China. [2]State Key Laboratory for Modification of Chemical Fibers and Polymer Materials, Center for Advanced Low-dimension Materials, College of Materials Science and Engineering, Donghua University, 201620 Shanghai, China. [3]School of Materials Science and Engineering, National Institute for Advanced Materials, Renewable Energy Conversion and Storage Center (RECAST), Nankai University, 300350 Tianjin, China. [4]N. D. Zelinsky Institute of Organic Chemistry, Russian Academy of Sciences, 119991 Moscow, Russia. [5]These authors contributed equally: Huazhe Liang, Xingqi Bi. ✉e-mail: zyao@nankai.edu.cn; yschen99@nankai.edu.cn

levels; (2) enhancing intermolecular charge transfer and intermolecular packings simultaneously; (3) improving molecular crystalline ordering and film morphologies; (4) contributing to superior charge carrier transport behaviors, suppressed charge recombination and upgraded photovoltaic parameters, etc[28–33]. In addition, theoretical studies have also revealed that halogenation on SMAs is inclined to enlarge the variation of dipole moments between ground and excited states of SMAs, thus minimizing molecular exciton binding energies and endowing with SMAs highly efficient exciton dissociation even driven by a quite small energy offset[9,34–36].

As regards different halogenations, fluorine and chlorine atoms featuring small atomic radii but quite large electronegativity have been verified to possess powerful abilities in tuning physicochemical and photovoltaic characteristics of SMAs without bringing unfavorable steric hindrances[27,28,30,37]. In addition, the possible halogen bonds (X···H) and noncovalent interactions (X···S, X···π, etc.) induced by F or Cl could be also in favor of compact and ordered molecular packing for SMAs, thus leading to an enhanced charge transfer/transport property[38–40]. In a similar fashion, bromine is expected to play a more crucial role in constructing efficient SMAs when taking its similar ability to effectively tune energy levels and absorptions of SMAs but also several unique advantages into consideration. First, the electronegativity decreases with molar mass increasing from F to Br (F, 4.0; Cl, 3.0; Br, 2.8) but the atom size increases (F, 0.071; Cl, 0.099; Br, 0.114 nm). The large and loose outmost electron cloud makes bromine easily polarized and thus may result in stronger intermolecular interactions through the efficient orbital overlap of π/p-electrons[41], which is expected to improve the charge transport properties of conjugated materials[35,42]. Second, bromide possesses stronger crystallinity than that of fluoride and chloride, which could induce a better molecular crystalline ordering[43–46]. Third, brominated compounds are easy to synthesize under a mild condition and relatively low cost compared to fluoride and chloride, at the same time, stable enough when applied in light-harvesting materials. Given that bromine possesses similar abilities in tuning energy levels and packings of SMAs but extra merits of easily synthesized and polarized, stronger crystallinity compared to its homomorphic fluorine and chlorine, it is quite surprising that very rare high-performance brominated SMAs have been reported thus far[32,35,41–44,47–53]. This may be ascribed to the following two reasons. First, the routine brominating on end units of SMAs will damage the highly effective intermolecular packing mode of "end unit to end unit" due to steric hindrance caused by the relatively large atomic radius of Br (0.114 nm)[42,43]. While this may not play a dominant role in ITIC-series SMAs with bulky side group grafted sp³ carbon sites[41,49], nevertheless, in the state-of-the-art Y-series SMAs, the favorable 3D intermolecular packing network of SMAs may be broken, especially considering that the π-π stacking distance of "end unit to end unit" is only 0.32-0.36 nm[32,54]. Second, brominated SMAs usually possess stronger crystallinity and greatly reduced solubility due to heavy atomic effects, which is not conducive to forming superior nanoscale film morphology[42,49,53]. As has been discussed above, every coin has two sides, just like the introduction of bromine on SMAs. However, in light of the unique merits of bromine, the record-breaking OSCs can be really expected if we can maximize the advantages of bromine while circumventing its weaknesses successfully.

Recently, an excellent molecular platform of CH-series SMAs has been established by our group, which maintains the current best molecular skeleton with a distinctive phenazine core as central units[21,28,29]. The conjugated extension of central units with respect to Y-series SMAs could not only lead to smaller exciton binding energies (see below), superior molecular packing modes and enhanced molecular crystalline ordering but also endow SMAs with more structural optimization possibilities by providing sufficient active sites on molecular skeletons[28]. A systemic investigation has revealed the favorable 3D molecular packing network of CH-series SMAs is formed through three equally crucial structural units of two end groups and one central group[21,29]. Therefore, it is no wonder that a remarkable change of intermolecular packings and even photovoltaic parameters can be observed just by a minor structural modification on the phenazine central unit[29]. This unique character of CH-series SMAs really makes us excited because the benefits of bromine could be fully exerted without damaging the favorable intermolecular packing mode of "end unit to end unit" if introducing bromine on central units rather than conventional end groups. Therefore, we believe that state-of-the-art OSCs can be expected if delicately brominating is conducted on such a unique molecular platform of CH-series SMAs.

Bearing these thoughts in mind, three CH-series SMAs (CH20, CH21 and CH22) are constructed with stepwise brominating on central units instead of conventional end units (Fig. 1a). The selected fluorinated end units could maintain tight molecular packing of "end unit to end unit". Whereas the advantage of easily polarizing and high crystallinity for bromide can be fully released simultaneously in spite of its undesired steric hindrance. As a result, bromine on central unit of SMAs regulates molecular packings and film morphology significantly, thus giving rise to enhanced molecular packing, improved molecular crystalline ordering, reduced exciton binding energy and enlarged relative dielectric constant. Benefitting from the facilitated charge separation/transport dynamics, PM6:CH22-based OSC affords the highest efficiency of 19.06% for brominated SMAs-based simple binary devices (Supplementary Table 1). More excitingly, due to the strong crystallinity of CH22, a record-breaking efficiency of 15.70% was reached when further thickening active layers to ~500 nm. Note that the excellent thick-film tolerance for CH22 is conducive to feasible OSCs production using large area printing. By exhibiting such a rare case of high-performance brominated SMA, our work highlights the great potential for achieving record-breaking OSCs along with excellent thick-film tolerance through delicately brominating on SMAs.

## Results
### Physicochemical properties
Three SMAs of CH20, CH21 and CH22 can be successfully synthesized according to our previously developed method[21] with an excellent yield in each step, while the synthetic and characterization details have been provided in the Supplementary Methods, Supplementary Figs. 1–25 and Supplementary Data 1. In general, a desirable A-D-A architecture can be indicated by plots of frontier orbital charge density differences (ΔQ) along molecular skeletons with a clear peak-valley-peak shape (Supplementary Fig. 26) for all three SMAs[55]. Note that SMAs with such an A-D-A architecture are expected to possess improved light harvesting and charge generation/transport dynamics, reduced energy losses and thus enlarged PCEs in resulting OSCs compared to other type molecules[1,2,38,56,57]. In good accordance with our discussions above, the isotropic polarizability of phenazine cores increases gradually with the introduction of bromine, being 146.28 without bromine, 166.26 with one bromine and 185.01 with two bromines (Fig. 1b). This tendency can be also found in the resulting SMAs, displaying stepwise enlarged polarizability from CH20 to CH22 (Supplementary Fig. 27). In addition, the polarizability of dibrominated phenazine is also larger than those of its difluorinated and dichlorinated analogs (Fig. 1b).

The largest dipole moments were also observed for both dibrominated phenazine and CH22 with respect to their counterparts (Supplementary Figs. 27 and 28). It is worth noting that CH22 contributes to a larger relative dielectric constant ($\varepsilon_r$) of 3.32 than that of 2.00 for CH20 and 2.35 for CH21 (Supplementary Fig. 29). This could be ascribed to both the enlarged molecular polarizability and dipole moment of CH22 comparing with those of CH20 and CH21. The increased $\varepsilon_r$ of organic semiconductors is conducive to achieve facilitated charge transfer/transport dynamics and thus gives rise to a better OSC with amplified FF and $J_{SC}$[58,59]. Furthermore, based on the

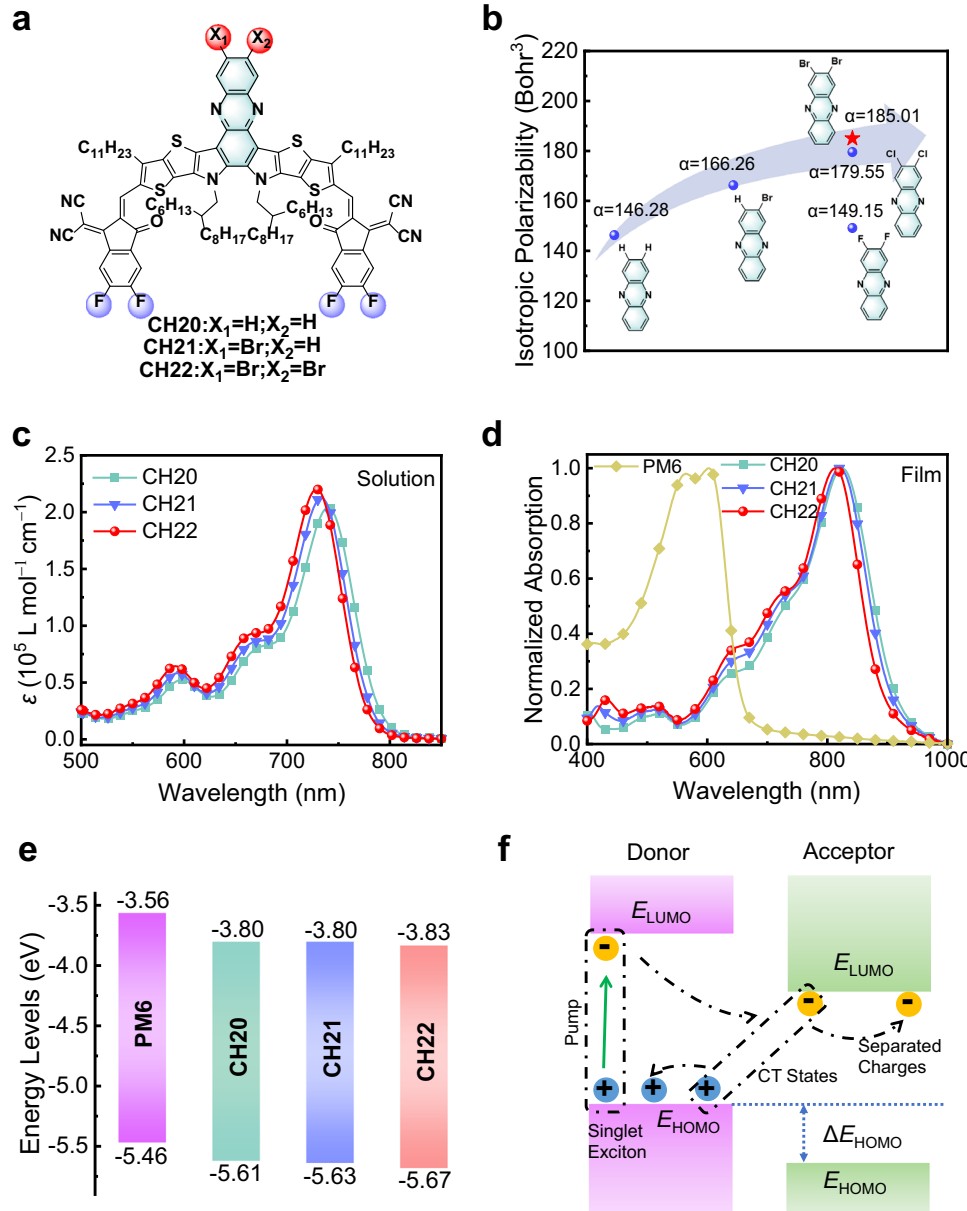

**Fig. 1 | Molecular structures and photophysical properties. a** Chemical structures of CH20, CH21 and CH22. **b** The isotropic polarizability. **c** Electron absorption spectra of CH20, CH21 and CH22 dissolved in dilute chloroform. **d** Normalized absorption spectra of neat films for PM6, CH20, CH21 and CH22. **e** Energy level diagram of PM6, CH20, CH21 and CH22 neat films derived from CVs. **f** Working mechanism diagram of free carrier generation at donor-acceptor interfaces.

differential scanning calorimetry (DSC) curves in Supplementary Fig. 30a, CH20, CH21 and CH22 exhibit exothermal peaks at 241.5, 253.9 and 286.8 °C, respectively, corresponding to a melting enthalpy ($\Delta H_m$) of 16.3 for CH20, 17.1 for CH21 and 23.3 J g$^{-1}$ for CH22. The larger $\Delta H_m$ of CH22 than those of CH20 and CH21 suggests its stronger molecular interactions[60], which will be discussed in detail below.

To evaluate the effects of central unit brominating on light-harvesting capacities, the electron absorption spectra of SMAs were recorded. As presented in Fig. 1c, the maximum absorption peaks ($\lambda_{max}$) of CH21 and CH22 dissolved in chloroform are at 733 and 729 nm, slightly hypochromic shift by 7 and 11 nm, respectively, compared to that of 740 nm for CH20. Moreover, CH21 and CH22 exhibit enlarged molar absorption coefficients of 2.13 ×10$^5$ and 2.20×10$^5$ L mol$^{-1}$ cm$^{-1}$ in comparison to that of 2.04×10$^5$ L mol$^{-1}$ cm$^{-1}$ for CH20, which might be ascribed to the suppressed spin prohibition of Q bands in CH21 and CH22[50]. An obvious red-shifting of $\lambda_{max}$ can be observed from solutions to solid films for all the three SMAs (Fig. 1d),

implying strong intermolecular interactions in solid states[61,62]. The HOMO and LUMO of CH20, CH21 and CH22 afforded by cyclic voltammetry (CV) measurements could be calculated as −5.61/−3.80 eV, −5.63/−3.80 eV and −5.67/−3.83 eV, respectively (Fig. 1e and Supplementary Fig. 31). Obviously, the gradually downshifted HOMOs should be ascribed to the introduction of electronegative bromine on central unit/donor of SMAs. Meanwhile, the energy gaps derived from CVs can be calculated as 1.36, 1.37 and 1.39 eV for CH20, CH21 and CH22, respectively. The gradually enlarged energy gaps agree well with the blue-shifted $\lambda_{onset}$ as mentioned above (Fig. 1d). Please note that the downshifted HOMO of CH22 will result in a relatively larger driving force for charge separation than CH20 and CH21 when the same donor is applied (Fig. 1f). The relative alignments of HOMO and LUMO agree well the tendency rendered by density functional theory (DFT) calculations (see Supplementary Note 1 and Supplementary Fig. 32).

As shown in Supplementary Fig. 30b, all three SMAs possess excellent thermal stability, indicated by a high decomposition

temperature of more than 330 °C. Moreover, all three SMAs also exhibit excellent photostability, indicated by their UV-vis spectra. As shown in Supplementary Figs. 33 and 34, the shape and intensity of absorption spectra display almost no changes after aging for over 550 h under one sun illumination[63]. Simultaneously, the photostability of CH22 was further confirmed by nuclear magnetic resonance (NMR) spectroscopy. There is no change before and after aging for over 450 h under one sun illumination (Supplementary Fig. 35), which suggests that the introduction of C-Br bonds does not decrease the photostability of acceptors[64]. The excellent thermal stability and photostability of these three SMAs can meet well the chemical stability requirement as light absorption materials. In order to make a clear comparison, the related physicochemical data of CH20, CH21 and CH22 have been summarized and presented in Supplementary Tables 2 and 3.

## Molecular packing in single crystals

In order to unveil the effects of central unit brominating on optimized molecular geometries and packing behaviors, single-crystal X-ray diffraction measurements of CH20, CH21 and CH22 were performed. By use of a slow solvent diffusion method, three single crystals featured with a beautiful metallic luster were afforded (see Supplementary Note 2). Their detailed X-ray parameters and checked files about the structure factors and structural output of the three single crystals can be found in Supplementary Table 4, Supplementary Figs. 36–38 and Supplementary Data 2–4. As presented in Fig. 2a, the hexyl decyl substitutions on SMAs were omitted for clear observation. All three SMAs exhibit a similar banana-curved and helical molecular geometry. Note that the distances of S...N between phenazine and bridged thiophene are ~3.3–3.4 Å for all the three acceptors, slightly smaller than the sum van der Waals radii (~3.55 Å when using the values of Alvarez[65]) of S and N, indicating the possible noncovalent interactions between S and N in these SMAs. This could also be supported by the reduced density gradient (RDG) analysis indicated by the light blue isosurfaces between S and N atoms (Supplementary Fig. 39)[66–68]. Note that

the effective noncovalent S...O = C and S...N secondary interactions further guarantee the relatively planar conjugated backbones[21,29,32]. Among them, CH21 and CH22 exhibit a slightly distorted skeleton with a larger dihedral angle of ~11° between two end groups than that of 3.59° for CH20, which should be caused by the bromine-induced steric hindrance on central units. From the overall view in Fig. 2b, all three crystals can be classified as triclinic systems, and more importantly, the favorable three-dimensional (3D) molecular packing networks can be well established. In addition, CH21 and CH22 possess rectangle-shaped voids of ~16.75 × 13.92 Å and ~16.66 × 14.03 Å, respectively, smaller than that of ~18.20 × 14.30 Å for CH20 with a rectangle-shaped void. It is worth mentioning that the favorable 3D molecular packing networks are usually constructed by both central and end units in CH-series SMAs and have been proven to be conducive to get superior charge separation/transport/recombination dynamics in OSCs[28].

The diverse 3D molecular packing networks for CH20, CH21 and CH22 should be on the basis of different intermolecular packing modes, which could be dramatically affected by even a very little structural modification (like different halogenations) on the central unit of CH-series SMAs[29]. Therefore, all the intermolecular packing modes with intermolecular potential over 70 kJ mol⁻¹ were extracted from single crystals of CH20, CH21 and CH22 (Fig. 3), and a detailed analysis was presented in Table 1. As regards CH20 without bromine on the central unit, three packing modes can be observed, which are assigned to dual end to central units ("dual E/C" mode), dual end to bridge units ("dual E/b" mode) and central to central units ("C/C" mode). Among them, the "dual E/C" mode with a small π...π stacking distance of 3.425 Å and an intermolecular potential of 190.5 kJ mol⁻¹ was only observed in CH-series SMAs thus far and played a crucially important role in establishing superior 3D packing networks[21,29]. More interestingly, the conventional packing mode of end-to-end units ("E/E" mode), which exists widely and even plays a highly important role in high-performance ITIC and Y-series SMAs[1,54,61,69,70], vanishes from sight in the crystal of CH20 along with the emergence of a never-observed but also effective "C/C" mode. The possible reason for forming "C/C"

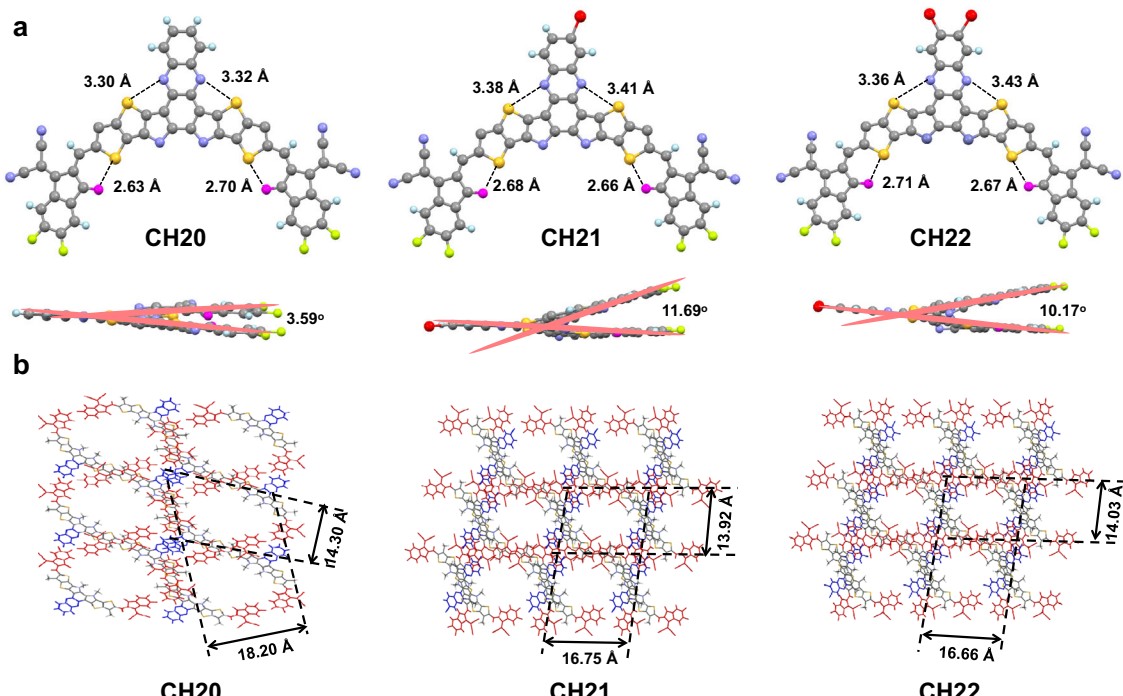

**Fig. 2 | Single-crystal structures of CH20, CH21 and CH22. a** Monomolecular single crystallographic structure of CH20, CH21 and CH22 in top- and side-view. **b** Single-crystal packing images on the top view of CH20, CH21 and CH22. Red, gray and blue colors highlight the end groups (E), bridge unit (b) and central unit (C), respectively.

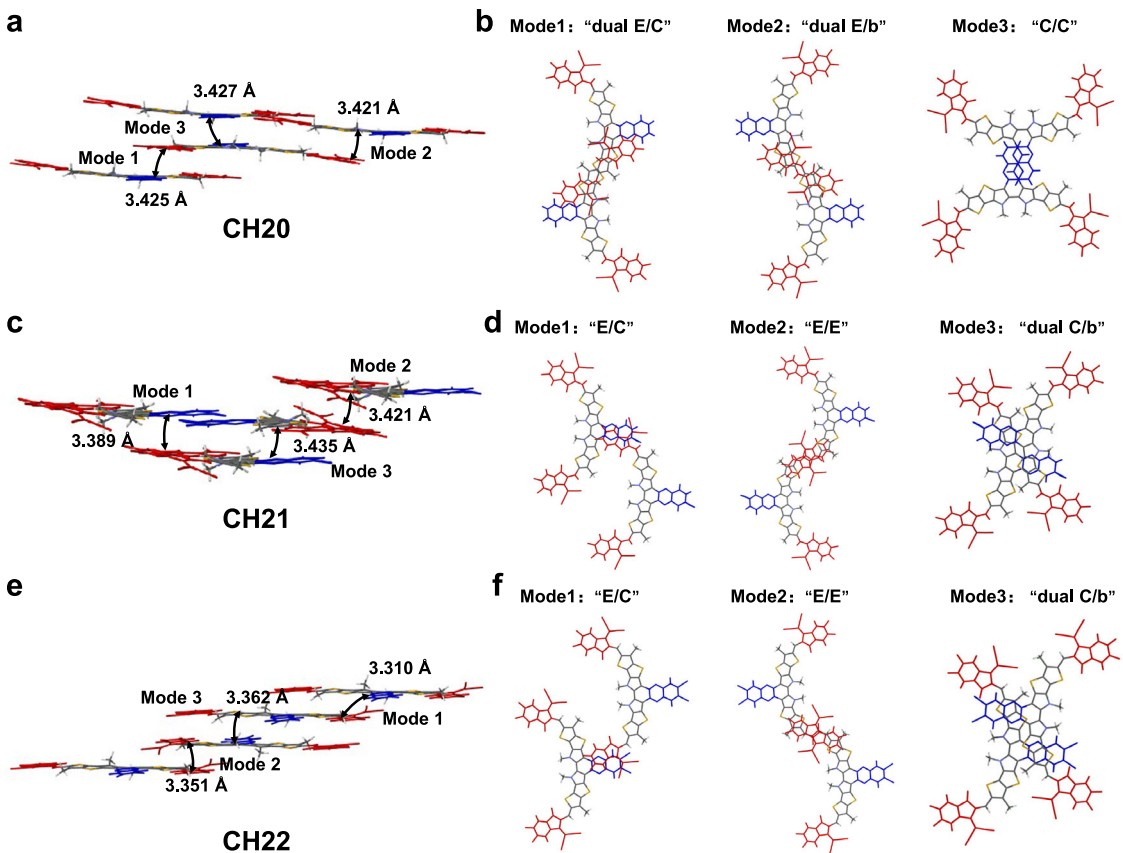

**Fig. 3 | Intermolecular packing modes of CH20, CH21 and CH22.** π-π stacking distances of interlayer involving with all the studied intermolecular packing modes of **a**, **b** CH20, **c**, **d** CH21 and **e**, **f** CH22, respectively. Red, gray and blue colors highlight the end groups (E), bridge unit (b) and central unit (C), respectively.

**Table 1 | Crystallographic and π-π interaction parameters of CH20, CH21 and CH22**

| Compound | Void sizes (shape) | Packing modes | $d_{π-π}$ [a] (Å) | Intermolecular potentials (kJ mol⁻¹) | $V_E$ [b] (meV) | $μ_{e,avg}$ [c] (10⁻⁴ cm² V⁻¹ s⁻¹) |
|---|---|---|---|---|---|---|
| CH20 | 18.20 × 14.30 Å | Mode 1 (Dual E/C) | 3.425 | 190.5 | 21.42 | 2.14 |
| | | Mode 2 (Dual E/b) | 3.427 | 162.5 | 5.41 | |
| | | Mode 3 (C/C) | 3.421 | 76.7 | 7.21 | |
| CH21 | 16.75 × 13.92 Å | Mode 1 (E/C) | 3.389 | 75.8 | 4.78 | 3.45 |
| | | Mode 2 (E/E) | 3.421 | 129.2 | 56.84 | |
| | | Mode 3 (Dual C/b) | 3.435 | 194.8 | 0.82 | |
| CH22 | 16.66 × 14.03 Å | Mode 1 (E/C) | 3.310 | 79.7 | 6.48 | 4.63 |
| | | Mode 2 (E/E) | 3.351 | 129.9 | 54.69 | |
| | | Mode 3 (Dual C/b) | 3.362 | 195.5 | 16.13 | |

[a]$d_{π-π}$ is the π-π distance of intermolecular packing modes of CH20, CH21 and CH22, which is consistent with that of Fig. 3.
[b]$V_E$ is the electron transfer integrals.
[c]$μ_e$ is the electron mobility of the corresponding neat films, which was measured by the SCLC method. The average $μ_e$ calculated from five independent devices.

packing mode in CH20 may be that the thermodynamically stable state of "C/C" packing is more preferential during the very slow crystallization process due to the obvious π-conjugated expansion of the central unit. However, with brominating on the central unit of CH20, the "C/C" packing mode disappears, and the highly effective "E/E" packing mode reemerges again with a large intermolecular potential of -129 kJ mol⁻¹. This should be ascribed to the relatively large steric hindrance of bromine on central units of CH21 and CH22, which significantly weakens the tendency for forming "C/C" packing mode and facilitates the formation of "E/E" packing in turn. Besides, another two distinctive packing modes of end-to-central units ("E/C" mode) and dual central to bridge units ("dual C/b" mode) could be also observed

in crystals of CH21 and CH22. Focusing on these three packing modes in CH20, CH21 and CH22, dibrominated CH22 shows a smaller average π···π distance ("E/C" mode: 3.310 Å, "E/E" mode: 3.351 Å, "dual C/b" mode: 3.362 Å) than those of CH20 ("dual E/C" mode: 3.425 Å, "dual E/b" mode: 3.427 Å, "C/C" mode: 3.421 Å) and CH21 ("E/C" mode: 3.389 Å, "E/E" mode: 3.421 Å, "dual C/b" mode: 3.435 Å). This may benefit from the more effective Br···S, Br···π or halogen bonding interactions[41,49] in the crystal of CH22 despite the relatively large steric hindrance of bromine. Note that the favorable molecular packings endow with CH22-based neat film an improved electron mobility ($μ_e$, Supplementary Fig. 40a, b) of 4.63 × 10⁻⁴ cm² V⁻¹ s⁻¹, compared to that of CH20 (2.14 × 10⁻⁴ cm² V⁻¹ s⁻¹) and CH21 (3.45 × 10⁻⁴ cm² V⁻¹ s⁻¹)[21,29,70].

Also, in order to unveil the change tendency of exciton binding energy ($E_b$), we took several single-crystal structures of CH-series SMAs to theoretical calculations and made a comparison with typical Y6 in the gas phase (Supplementary Fig. 41 and Supplementary Table 5). As a result, the $E_b$ of CH-series SMAs is much lower than that of Y6 and, more importantly, also decreases with halogenation on central units, such as from Y6 to CH20 and CH22. It is well known that $E_b$ is influenced by not only molecular structures but also solid-state polarization effects and intermolecular electronic interactions[34,71]. Therefore, the calculated values of $E_b$ shown in Supplementary Table 5 are dramatically larger than that observed in the solid phase based on the temperature-varying photoluminescence (PL) spectra[72]. As displayed in Fig. 4, all the CH-series SMAs possess a greatly smaller experimental $E_b$ (~100 meV) than that of 226 meV for the Y6 film, indicating the advantages of expanded conjugation of central units. More importantly, $E_b$ could decrease with halogenation on the central unit of CH-series SMAs, making it high potential for achieving a much smaller $E_b$ (even comparable to inorganic semiconductors) if further halogenation on the central unit is performed to delicately tune both molecular structures and intermolecular packing modes.

To sum up, CH20 with a non-brominated central unit was observed with a strong intermolecular packing mode of "C/C" due to the great π-conjugated extension of the central unit, thus suppressing the formation of the conventional packing mode of "E/E". After brominating on the central unit, the "C/C" packing mode disappears, and the highly effective "E/E" packing mode reemerges, indicating that the brominating on the central unit of SMAs has a profound effect on intermolecular packing behaviors. More importantly, the "E/E" packing mode has been proven to play a very crucial role in establishing the favorable 3D molecular packing networks of SMAs and thus leads to more efficient charge transport. Therefore, on such an excellent molecular platform of CH-series SMAs, brominating on a central unit rather than an end unit could better balance the different intermolecular packing modes and afford a much smaller $E_b$, thus max-

imizing the advantages of bromination (like high crystallinity, efficient halogen bonding, etc.) while circumventing its adverse effects on molecular packings.

## Photovoltaic performances

The diverse intermolecular packing modes of these SMAs should lead to much different photovoltaic performances of resulting OSCs in theory. Thus, OSCs featured with a device architecture of ITO/PEDOT:PSS/active layer/PNDIT-F3N/Ag were fabricated, in which a polymeric donor PM6[4] was blended with CH20, CH21 and CH22 to compose active layers because of its matched energy levels and absorptions (Supplementary Fig. 42). Detailed device fabrication and characterization procedures were provided in Supplementary Information (Supplementary Tables 6–8). The current density-voltage ($J$-$V$) characteristics of the optimal OSCs are shown in Fig. 5a and illustrated in Table 2. For CH20-based binary OSCs, a PCE of 16.79% with a $V_{OC}$ of 0.881 V, a $J_{SC}$ of 25.44 mA cm$^{-2}$ and an FF of 74.92% is delivered. Benefiting from the optimized molecular packings tuned by brominating on central units, a greatly improved PCE of 18.12% is yielded by a CH21-based device accompanied by a $V_{OC}$ of 0.873 V, a $J_{SC}$ of 26.57 mA cm$^{-2}$ and an FF of 78.13%. More excitingly, a champion PCE of 19.06% is further generated by CH22-based OSCs featured with a $V_{OC}$ of 0.884 V, a $J_{SC}$ of 26.74 mA cm$^{-2}$ and an outstanding FF of 80.62%, which has been ranked among the first-class OSCs thus far[10,11] and also reached the best PCE for CH-series SMAs-based OSCs[21,28]. The statistic efficiency analysis for 15 independent OSCs (see the detailed device parameters in Supplementary Tables 9–11) was inset in Fig. 5a, and the average performance from CH20 to CH22 also shows an enlarged tendency. The corresponding external quantum efficiency (EQE) plots of CH20-, CH21- and CH22-based devices were displayed in Fig. 5b. Note that the integrated current densities derived from EQE plots are 24.69, 25.99, and 26.17 mA cm$^{-2}$, respectively, which matched well with those afforded by $J$-$V$ tests (within 3% error). Although the gradually

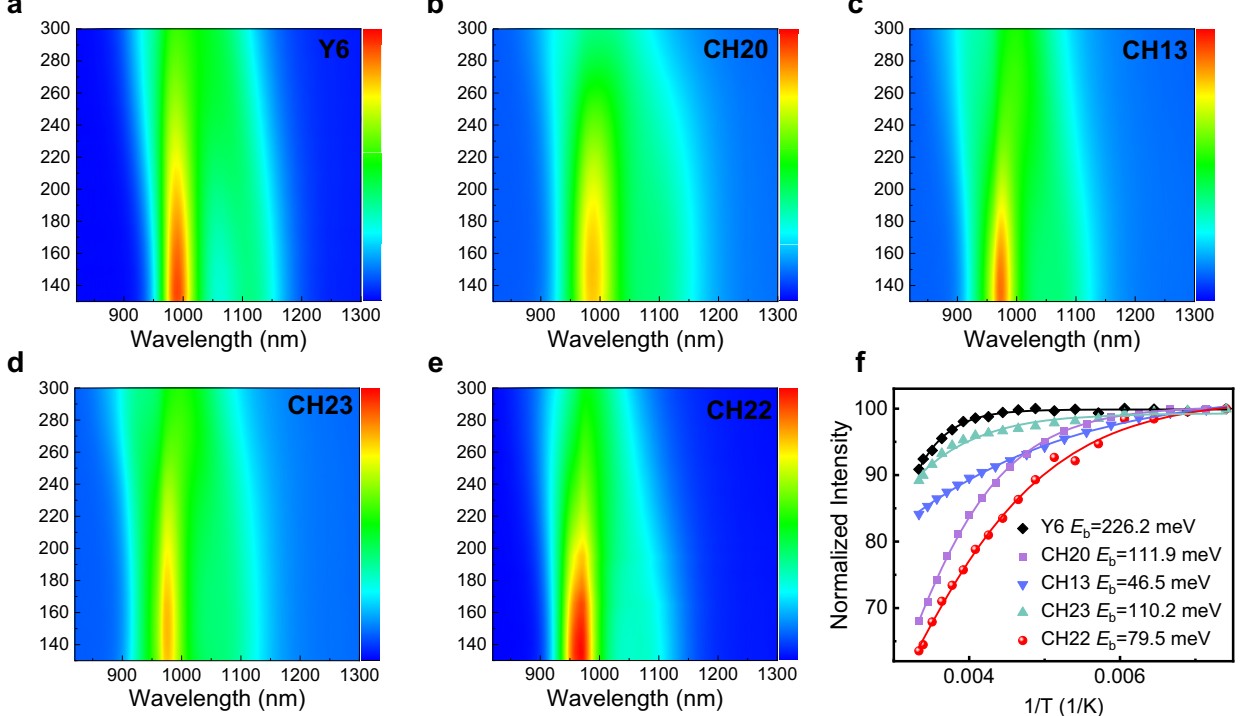

**Fig. 4 | The experimental exciton binding energy of SMAs.** The temperature-dependent PL spectra of **a** Y6, **b** CH20, **c** CH13, **d** CH23 and **e** CH22 neat films. **f** Temperature-dependent data of integrated intensity.

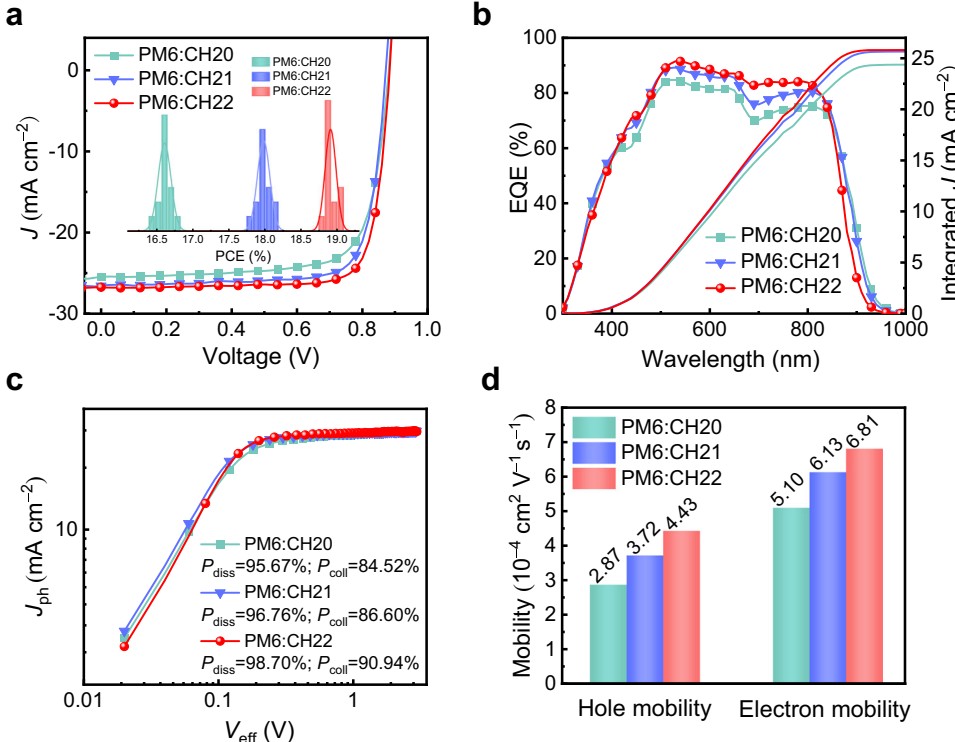

**Fig. 5 | Photovoltaic performances of the optimized OSCs. a** Current density-voltage curves of PM6:CH20, PM6:CH21 and PM6:CH22-based OSCs; the inset shows a statistical analysis of device efficiency. **b** The EQE plots and integrated $J_{SC}$ curves. **c** $J_{ph}$ versus $V_{eff}$ curves of PM6:CH20, PM6:CH21 and PM6:CH22-based OSCs. **d** Histograms of the hole and electron mobility of blended films.

**Table 2 | Summary of device parameters for optimized OSCs[a]**

| Active layers | $V_{OC}$ (V) | $J_{SC}$ (mA cm$^{-2}$) | Cal. $J_{SC}$[b] (mA cm$^{-2}$) | FF (%) | PCE (%) |
|---|---|---|---|---|---|
| PM6:CH20 | 0.881 (0.881 ± 0.003) | 25.44 (25.27 ± 0.17) | 24.69 | 74.92 (74.62 ± 0.30) | 16.79 (16.61 ± 0.09) |
| PM6:CH21 | 0.873 (0.872 ± 0.002) | 26.57 (26.57 ± 0.07) | 25.99 | 78.13 (77.56 ± 0.36) | 18.12 (17.98 ± 0.09) |
| PM6:CH22 | 0.884 (0.884 ± 0.002) | 26.74 (26.62 ± 0.13) | 26.17 | 80.62 (80.34 ± 0.24) | 19.06 (18.91 ± 0.08) |

[a]The average parameters afforded by 15 independent devices (Supplementary Tables 9–11).
[b]Current densities calculated from EQE curves.

blue-shifted absorption from CH20 to CH22 renders a stepwise shrunken EQEs spectra, the overall higher EQE responses in the range of 450–850 nm for CH22 contribute to the largest integrated current densities. Note that the varying EQE values are involved with multiple factors, such as the efficiencies of light harvest, exciton dissociation, charge transport, etc. Herein, the more effective exciton dissociation caused by a relatively large driving force (Fig. 1f) and facilitated charge transport induced by superior nanoscale film morphologies (see the detailed discussions below) for CH22 should account for its best EQEs and integrated current density. In addition, the significantly improved FF for CH22-based OSCs should be caused by facilitated charge transport. It is also worth noting that PCEs of PM6:CH22-based OSCs could be maintained above 96% and ~85% compared to its initial PCEs after 1500 h under room temperature and 400 h under heat treatment at 65 °C (Supplementary Fig. 43), respectively. The good storage and thermal stabilities of PM6:CH22 system highlight its potential for commercialization in large-scale production of high-performance OSCs.

In order to verify the unique advantages of brominating on the central unit rather than end groups, as we have discussed above, we further constructed the SMAs of CH22-6Br with di-bromination on both central unit and end groups. As shown in Supplementary Fig. 44 and Supplementary Table 12, CH22-6Br only achieved a PCE of 17.00%

with significantly decreased $J_{SC}$ and FF compared to those of CH22, which may be caused by not only the excessive molecular aggregation but also the less compact π-π stacking of molecules due to the relatively larger steric hindrance of bromine than its homomorphic fluorine or chlorine. The excessive molecular aggregation of CH22-6Br can be confirmed by the relatively larger fiber size (12.7 nm for PM6:CH22-6Br, 11.9 nm for PM6:CH22) based on a statistical analysis of atomic force microscopy (AFM) phase images (Supplementary Fig. 45) and slightly greater crystal coherence length (CCL) of 23.37 Å in CH22-6Br neat film estimated from GIWAXS (Supplementary Fig. 46 and Supplementary Table 13) comparing to that of CH22 (22.53 Å). Moreover, the slightly larger π-π stacking distances for CH22-6Br (3.74 and 3.70 Å for neat and blended films) can be estimated from GIWAXS compared to that of CH22 (3.66 and 3.63 Å for neat and blended films), which may be caused by the relatively larger steric hindrance of bromine.

To further understand the fundamental reasons for $J_{SC}$ and FF enlargement in CH21- and CH22-based devices with respect to that of CH20, both the efficiencies for exciton dissociation ($P_{diss}$) and charge collection ($P_{coll}$) were first investigated (Fig. 5c). Compared with that of 95.67%/84.52% for CH20-based device, the values of $P_{diss}/P_{coll}$ can be estimated as 96.76%/86.60% and 98.70%/90.94% for CH21- and CH22-based devices, respectively. The improved $P_{diss}$ should be ascribed to

the downshifted HOMO energy levels of CH21 and CH22, which will theoretically lead to a larger driving force for exciton splitting[9,34]. In addition, the enlarged $P_{coll}$ for CH21 and CH22-based devices should be attributed to superior charge transportation dynamics[35]. This can be manifested by the smaller charge extraction time of 0.32 μs for CH22 and 0.39 μs for CH21-based devices compared to that of 0.48 μs for CH20-based one (see the transient photocurrent decay curves in Supplementary Fig. 47). As we have discussed above, the greater polarizability and stronger delocalization effect with the introduction of bromine on SMAs are expected to result in stronger intermolecular interactions and increased charge mobility. Therefore, the electron/hole mobilities ($\mu_e/\mu_h$) of PM6:CH20-, PM6:CH21- and PM6:CH22-based devices were evaluated by employing the space-charge limited current (SCLC) method, which can be determined to be 5.10/2.87, 6.13/3.72 and 6.81/4.43 × $10^{-4}$ cm$^2$ V$^{-1}$ s$^{-1}$ corresponding to $\mu_e/\mu_h$ ratios of 1.78, 1.65 and 1.54, respectively (Fig. 5d and Supplementary Fig. 40c, d). Such results indicate that the CH22 system with the strongest polarizability and largest dipole moment among these three SMAs could effectively optimize the charge separation and transport dynamics to further improve $J_{SC}$ and FF in corresponding devices.

## Morphology analysis

As it has been widely discussed, the morphology of blended films in OSCs plays even a dominant role in charge transport behavior and photovoltaic performance[11,31]. Therefore, we implemented the AFM and transmission electron microscopy (TEM) measurements (Supplementary Figs. 48 and 49) to unveil the nanoscale morphology of blended films. As shown in Supplementary Fig. 48a, all the blended films possess a uniform and smooth surface. The values of root-mean-square (RMS) roughness of PM6:CH20 (1.02 nm), PM6:CH21 (1.13 nm) and PM6:CH22 (1.22 nm) increased with the gradual introduction of bromine on the central unit due to the enhanced molecular crystallinity from CH20 to CH22. In view of the phase images (Supplementary Fig. 48b), fiber-like domains with proper sizes were observed, which has been proven to effectively facilitate charge transport in OSCs[19]. Based on a statistical analysis of nanofiber size in Supplementary Fig. 48c, d, a slightly but gradually increased fiber size can be observed with 10.7 for CH20, 11.5 for CH21 and 11.9 nm for CH22, suggesting that brominating on central units could enhance molecular crystallinity and fine-tune phase domain sizes. Meanwhile, TEM images (Supplementary Fig. 49) further corroborate the fibrillar network structure in all blended films, which is conducive to achieving high charge mobility and avail charge transport. Note that the nanofiber size is related to the miscibility between PM6 and SMAs; thereby, contact angles and derived Flory-Huggins interaction parameters ($\chi$)[73,74] for the corresponding donor and acceptor were further evaluated. As shown in Supplementary Fig. 50 and Supplementary Table 14, $\chi_{D:A}$ for PM6:CH22 (0.74) and PM6:CH21 (0.63) are larger than that of PM6:CH20 (0.46), indicating lower D/A miscibility after brominating on the central unit of SMAs[13]. This may contribute to the higher domain purity and larger nanofiber size, which is in good accordance with results from AFM and TEM images.

Then, the effect of brominating on a central unit of SMAs on molecular packing and orientation in neat and blended films was studied by employing GIWAXS. The 2D-GIWAXS patterns and line-cut profiles of CH20, CH21 and CH22 neat films are presented in Fig. 6, and corresponding parameters are summarized in Supplementary Table 15. All the neat films for CH20, CH21 and CH22 exhibit pronounced (010) diffraction peaks located at 1.663, 1.690 and 1.718 Å$^{-1}$ in out-of-plane (OOP) direction and sharp (100) diffraction peaks at 0.423, 0.418, 0.410 Å$^{-1}$ in in-plane (IP) direction, respectively, indicating the preferential face-on orientations. Note that CH22 displays the more prominent (010) diffraction peaks with slightly shifted to higher q region, assigned to a shorter π-π stacking distance of 3.66 Å than those of 3.78 Å for CH20 and 3.72 Å for CH21, which is in good

accordance with the analysis of single crystals (Supplementary Fig. 51). Meanwhile, a comparable but slightly larger CCL of 22.53 Å can be observed for CH22 neat film comparing to those of 21.10 Å for CH20 and 21.18 Å for CH21, indicating the enhanced molecular packing ordering. After blending with the PM6 donor, the sharp (010) diffraction peaks and (100) diffraction peaks can still be observed in OOP and IP directions, respectively, suggesting that the desirable face-on orientation could be well maintained. As it is summarized in Supplementary Table 15, the (010) diffraction peaks in the OOP direction for three blended films are located at 1.713, 1.718 and 1.733 Å$^{-1}$, assigned to the π-π stacking distances of 3.67, 3.66 and 3.63 Å, respectively. Meanwhile, CCLs of (010) diffraction peaks were estimated to be 22.00, 22.09 and 22.44 Å for PM6:CH20, PM6:CH21 and PM6:CH22 blended films, respectively. In general, the more compact π-π stacking and slightly enhanced molecular packing order for CH22 films should be contributed to the enhanced molecular crystallinity and response for the superior charge transport behaviors and even enlarged $J_{SC}$ and FF values in CH22-based OSCs[75]. Moreover, the Urbach energy ($E_u$) values were calculated to be 20.7, 20.1 and 19.3 meV for CH20-, CH21- and CH22-based devices, respectively, by fitting the Fourier transform photocurrent spectrometer EQE (FTPS-EQE) (Supplementary Fig. 52). The smaller $E_u$ of PM6:CH22 also suggests the more ordered molecular packing, which agrees well with its slightly larger CCL[76]. All the results above manifested that the delicately brominating on central units of SMAs could give rise to enhanced molecular crystallinity, more compact intermolecular packings and crystalline orders, thus rendering more efficient charge transport behaviors and better photovoltaic performance.

## Energy loss analysis

To study the influence of central unit brominating on energy loss ($E_{loss}$) of SMAs, all three SMAs devices were prepared. The total $E_{loss}$ for PM6:CH20, PM6:CH21 and PM6:CH22 systems are 0.495, 0.525 and 0.530 eV, respectively (see Supplementary Note 3 and Supplementary Figs. 53 and 54 for the detailed calculations). Note that CH21 and CH22 systems with brominating on central units afford the slightly larger $E_{loss}$ (Fig. 7, Supplementary Table 16), especially non-radiative recombination loss ($\Delta V_{nr}$). By fitting the low-energy region of highly sensitive EQE and EL spectra (Supplementary Fig. 54), the energy level ($E_{CT}$) of charge transfer (CT) state can be determined to be 1.35, 1.35 and 1.37 eV for PM6:CH20, PM6:CH21, PM6:CH22 blended films, respectively, giving rise to energy offsets ($\Delta E_{CT}$) between local exciton state (LE) and CT state of 0.026, 0.043 and 0.044 eV. With such a CT state energy near to that of LE, the highly possible hybridization of LE and CT states should be taken into consideration[77,78]. Note that the smaller $\Delta E_{CT}$ for CH20 than those of CH21 and CH22 may result in more effective hybridization of LE and CT states, thus enhancing the luminescence of the CT state through the "intensity borrowing" mechanism[29,78] and further suppressing the non-radiative recombination rate of CT states. This may be the reason that CH20-based OSC exhibits a reduced non-radiative recombination loss compared to that of CH21 and CH22-based ones. In addition, the relatively larger electron reorganization energies of 147.23 for CH21 and 148.87 for CH22 than that of 144.59 meV for CH20 can be observed (Supplementary Fig. 55), which should be caused by the easily polarizing property of bromine and also partially account for the larger $E_{loss}$ for CH21 and CH22-based OSCs. This is important as it is well known that a small energy loss should be rationally pursued on the premise of efficient charge transfer/transport dynamics. Therefore, a well-balanced trade-off between $J_{SC}$ and $V_{OC}$ is highly desirable, especially in the state-of-the-art OSCs. As displayed in Fig. 7c, the CH22-based device achieved the most compromise between tangle $J_{SC}$ and $V_{OC}$ in spite of its relatively larger energy loss with respect to that of CH20, thus rendering the best FF to reach the optimum efficiency.

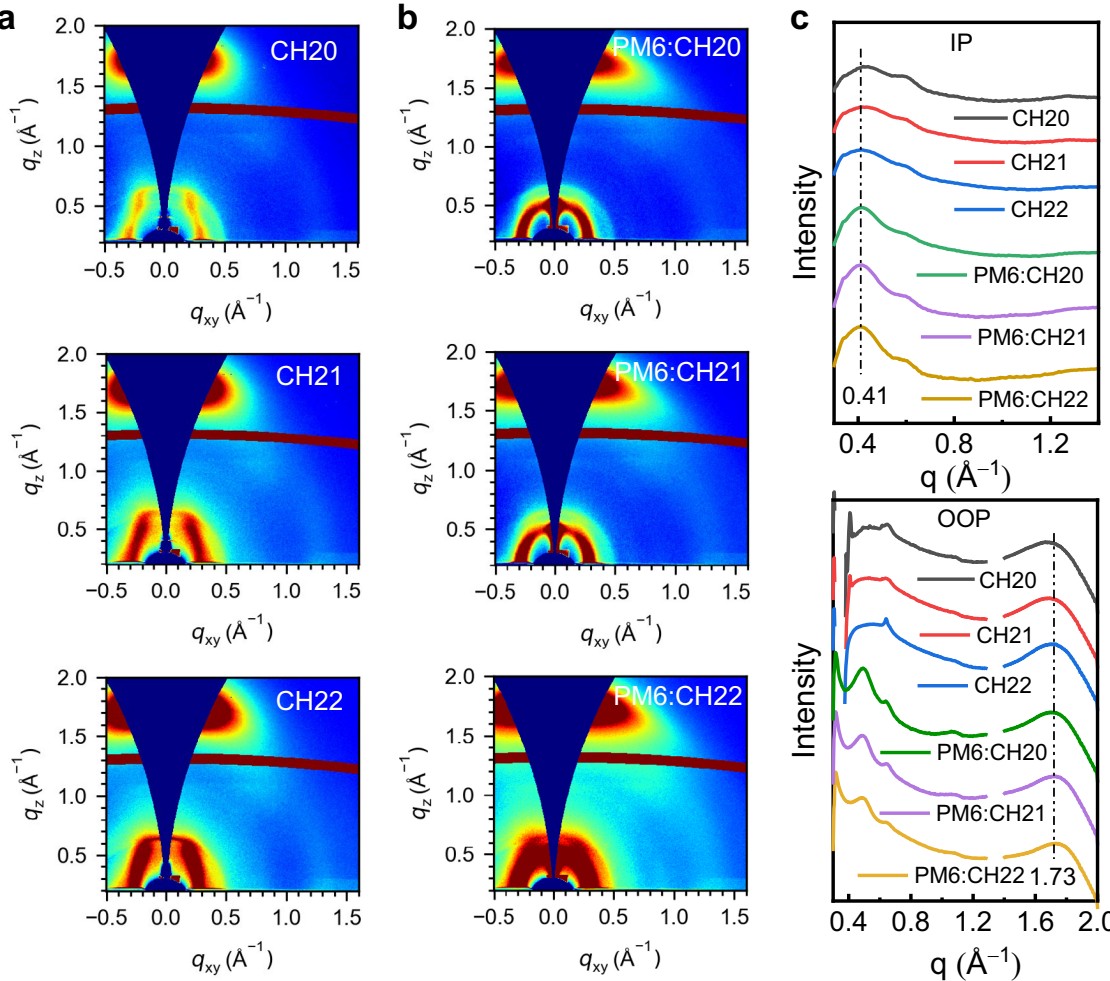

**Fig. 6 | GIWAXS characterization of the neat films and blended films. a** 2D GIWAXS patterns of CH20, CH21 and CH22 neat films and **b** optimized PM6:CH20, PM6:CH21 and PM6:CH22 blended films. **c** The corresponding in-plane (IP) and out-of-plane (OOP) extracted line-cut profiles of CH20, CH21 and CH22-based neat and blended films.

## Thick-film device performance

As we all know, the vast majority of high-performance OSCs are obtained with a layer thickness of ~100 nm and PCEs will decrease drastically with increasing thickness of active layers, which greatly limits their application in large area printing process using the most promising technology of roll to roll[79]. Thus, it is very important to develop high-performance OSCs with good tolerance of active layer thickness. It is well known that light-harvesting materials with high crystallinity and mobility are beneficial to obtain better performance with thick films[80]. Therefore, CH22 is expected to gain a satisfactory performance in thick-film devices in light of its compact molecular packings and high crystallinity. Bearing these thoughts in mind, PM6:CH22 devices with different active layer thicknesses (~200, ~300, ~400 and ~500 nm) were fabricated. The obtained $J$-$V$ characteristics and EQE plots were presented in Fig. 7e, f, and the detailed parameters are listed in Supplementary Table 17. With film thickness increasing, the $J_{SC}$ of CH22-based OSCs improves gradually, but FF declines greatly. As a result, an eventual PCE of 15.70% is afforded by applying a ~500 nm film thickness of CH22-based OSC, demonstrating the highest PCE for OSCs with active layer thickness up to 500 nm (Supplementary Table 18). Note that the greatly decreased FF may be caused by the more trap-assisted recombination in thicker-film devices, suggested by increased S/($kT/q$) values from 1.08 to 1.26 and reduced $P_{coll}$ from 90.94 to 80.68% for OSCs with a film thickness from 100 to 500 nm (Supplementary Fig. 56).

## Discussion

Three A-D-A type SMAs (CH20, CH21 and CH22) have been constructed with delicately brominating on the central unit rather than conventional end groups. In this way, the unique advantages of bromides, like high crystallinity, easily polarizing, efficient halogen bonding interaction, etc., are maximized while circumventing the adverse effect of large steric hindrance on molecular packing, especially for the "E/E" packing mode. A systemic investigation has revealed that brominating on the central unit of CH-series SMAs could endow with CH22 increased polarizability and enlarged dipole moment, and thus improved relative dielectric constant. In addition, the large atomic radius of bromine on the central unit of SMAs transforms the intermolecular packing mode from "C/C" to the more effective "E/E" mode, leading to a more compact and ordered intermolecular packing and also superior 3D molecular packing network for CH22. More excitingly, the temperature-dependent PL measurement unveils that CH-series SMAs not only possess greatly reduced $E_b$ with respect to the state-of-the-art Y6 but also make it high potential for achieving a much smaller $E_b$ even comparable to inorganic semiconductors if further halogenation on the central unit is performed to delicately tuning both molecular structures and interocular packing modes. As a result, due to the facilitated charge separation/transport dynamics, PM6:CH22-based binary OSCs eventually render the highest efficiency of 19.06% for brominated SMA-based devices, along with good stability. Moreover, a record-breaking efficiency of 15.70% was also achieved when further

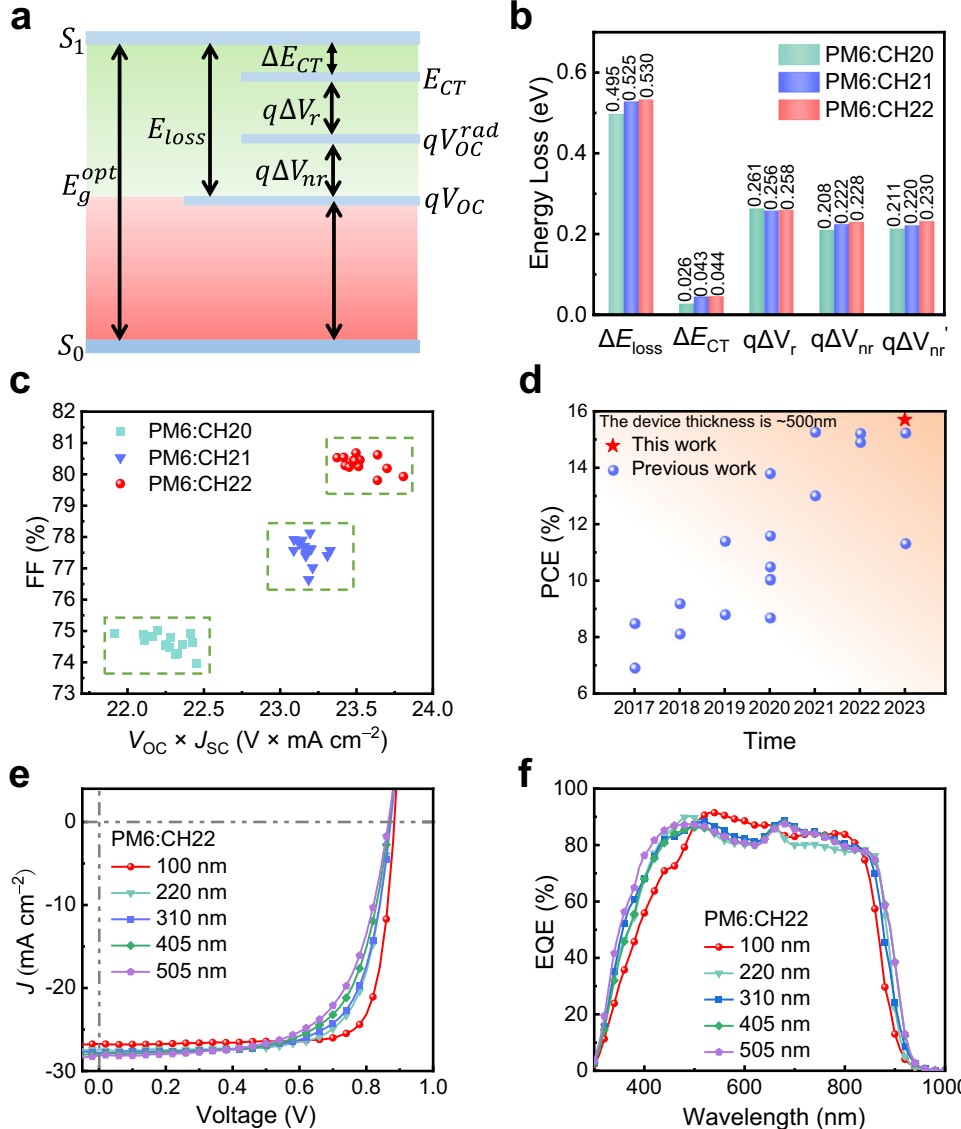

**Fig. 7 | Energy loss analysis of the optimized OSCs and the thick-film device performance of PM6:CH22. a** A schematic diagram of $E_{loss}$-related parameters. **b** Statistical diagram of detailed $E_{loss}$ of CH20, CH21 and CH22-based devices. **c** $V_{OC} \times J_{SC}$ vs FF of CH20, CH21 and CH22-based 15 independently measured OSCs. **d** A summary of the PCE for thick-film OSCs with ~500 nm. **e** Current density-voltage curves of PM6:CH22-based OSCs with different thicknesses (100 nm, 220 nm, 310 nm, 405 nm, and 505 nm). **f** The corresponding EQE spectra.

thickening active layers up to 500 nm, which is highly important to the large-scale production of OSCs. By developing such a rare case of high-performance brominated SMA, our work exhibits the great potential for achieving record-breaking OSCs through delicately brominating on SMAs and will stimulate further exploration of light-harvesting materials containing bromines.

## Methods

### Materials
The PM6 donor and IC-2F were purchased from Organtec Ltd. All the other reagents were purchased from commercial suppliers and used directly without further purification unless noted specially. The synthetic routes to acceptors of CH20, CH21 and CH22 and detailed synthesis processes were illustrated and described in Supplementary Information.

### Device fabrication
The conventional devices studied herein were featured with an architecture of ITO/PEDOT:PSS/PM6:SMAs/PNDIT-F3N/Ag. As regards the

device fabrication, the ITO glass was thoroughly cleaned with detergent water, deionized water, acetone and isopropyl alcohol in turn, by using an ultrasonic bath for 15 min and sequentially dried with a nitrogen purge. Then, the cleaned ITO glasses will be further treated with UV exposure for 15 min in a UV-ozone chamber (Jelight Company) before spin-coating (4300 rpm for 20 s) a thin layer of poly(3,4-ethylene dioxythiophene):poly(styrene sulfonate) (PEDOT:PSS, Baytron PVP Al 4083). Thereafter, under ambient conditions, the resulting PEDOT:PSS films were baked for 20 min at 150 °C to form a compact film and further transferred to a nitrogen-filled glovebox rapidly. Then PM6:CH20 and PM6:CH21 mixtures (D:A weight ratio = 1:1.2; a total concentration of 13.2 mg mL⁻¹) were dissolved in chloroform (CF) completely with 0.4% 1-chloronaphthalene (CN) as an additive. PM6:CH22 mixtures with different concentrations (D:A ratio = 1:1.2; 13.2 mg mL⁻¹ for 100 nm, 22 mg mL⁻¹ for 220 nm, 28.6 mg mL⁻¹ for 310 nm, 33 mg mL⁻¹ for 405 nm, 37.4 mg mL⁻¹ for 505 nm) were dissolved in CF completely with 0.4% 1-CN. The resulting solutions were stirred at room temperature for 4 h before spin-casting onto PEDOT:PSS layers at 1800 rpm for 30 s. Subsequently, the afforded films

were annealed at 80 °C for 5 min in a nitrogen-filled glovebox. After that, the PNDIT-F3N layer was spin-coated on the active layers at 3000 rpm for 20 s, using a methanol solution containing 0.5% v/v glacial acetic acid at the concentration of 1 mg mL$^{-1}$. Finally, an Ag back electrode with ~150 nm thickness was deposited under $2 \times 10^{-6}$ Pa through evaporation. The light active area of the device was ~4.1 mm$^2$ and the area of mask was ~2.58 mm$^2$ in our lab, which were determined using an optical profilometer (PSM-1000). The device's illuminated area was $7 \times 7$ cm$^2$ during the *J-V* measurements.

## Characterization of the OSCs

The *J-V* measurements were carried out by employing a solar simulator (SS-F5-3A, Enli Technology, xenon lamp, filter model AMFG2.0) with the standard AM 1.5 G spectra (100 mW cm$^{-2}$). The light density of the solar simulator was calibrated by a standard Si solar cell (made by Enli Technology Co., Ltd.), and the calibrated report could be traced to NREL. The spectral between reference cell and devices could match well during the test (within 3% errors). For the current-voltage tests under forward direction, the scan speed and dwell time are 0.02 V/s and 1 ms, respectively. No pretreatments (light soaking or holding cell at a bias) were required before *J-V* testing, and all the measurements were conducted in a nitrogen-filled glovebox at room temperature (ca. 25 °C) without attaching any antireflection coating on the incident plane of solar cells. Note that there is no hysteresis or other unusual behaviors during the measurements of solar cells. Devices for storage stability test were carried out in a glovebox at room temperature filled with ultrahigh purity nitrogen without encapsulation in dark conditions, and thermal stability test were carried out on a hot plate (WH200D-1K, WIGGENS, Germany) with a continuous thermal stress of 65 °C with nitrogen in a glovebox. The photostability measurements were carried out under continuous illumination provided by a LED light source (MT-PV-16, TIANJIN METO). The used spectrum was shown in Supplementary Fig. 33. The EQE plots were recorded by using a QE-R Solar Cell Spectral Response Measurement System (Enli Technology Co., Ltd.). Additional information for instruments and measurements used in this work were listed in Supplementary Note 3.

## Reporting summary

Further information on research design is available in the Nature Portfolio Reporting Summary linked to this article.

## Data availability

All data generated or analyzed in this study have been included in this published article and its Supplementary Information. Crystallographic data for the structures reported in this study have been deposited at the Cambridge Crystallographic Data Centre (CCDC) under deposition numbers CCDC 2244842 (CH20), 2244846 (CH21) and 2244848 (CH22). Copies of the data can be obtained free of charge via https://www.ccdc.cam.ac.uk/structures/. Source data are provided with this paper.

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

## Acknowledgements

The authors gratefully acknowledge the financial support from the Ministry of Science and Technology of the People's Republic of China (National Key R&D Program of China, 2022YFB4200400 (Y.C.), 2019YFA0705900 (X.W.)) and National Natural Science Foundation of China (21935007 (Y.C.), 52025033 (X.W.), 51873089 (H.Z.)), Tianjin city (22JCQNJC00530 (Z.Y.)) and Haihe Laboratory of Sustainable Chemical Transformations. The authors gratefully acknowledge the cooperation of the beamline scientists at BSRF-1W1A beamline.

## Author contributions

H.L. conducted the experiments and wrote the original paper. X.B. synthesized materials and grew the related single crystals. G.L., B.K. and T.H. carried out the theoretical computation. H.C., H.Z., and C.L. solved and analyzed the single-crystal structures. Y.Z. and W.F. performed the EL and sEQE experiments. Y.L. and Z.M. analyzed the data. K.M. performed the space-charge-limited current measurements and analyzed the data. O.A.R. and X.W. helped to analyze the data and revise the manuscript. Z.Y. and Y.C. supervised and directed this project. All authors discussed the results and commented on the manuscript.

## Competing interests
The authors declare no competing interests.
