## [Peer Review File · Nature Communications]

A Rare Case of Brominated Small Molecule Acceptors for High-Efficiency Organic Solar CellsREVIEWER COMMENTS

Reviewer #1 (Remarks to the Author):

Chen and co-authors constructed three A-D-A type NFAs with brominating on central unit instead of end groups. Therefore, the advantages of bromine can be maximized whilst circumventing its adverse effects of large steric hindrance on molecular packing, especially for “end unit to end unit” packing mode. By keeping a superior 3D molecular packing network, PM6:CH22-based binary OSCs eventually reach the highest efficiency of 19.06% for brominated NFAs-based devices along with a good stability. Moreover, an excellent efficiency of 15.70% was also achieved when further thickening active layers up to 500 nm. The authors indeed exhibit the great potentials for achieving high-performance OSCs through further brominating on NFAs, by developing such a rare case of brominated NFA. In addition, the manuscript has been well organized and the resulting conclusions are quite convincing at the current stage. Based on these considerations above, I recommended its publication in Nature Communications after a minor revision. Several suggestions are following:

1. In this text, the highest efficiency of 19.06% for brominated NFAs has been reported based on PM6:CH22. Comparisons with other brominated OSCs should be also mentioned (perhaps in an additional Table).
2. Compared with PM6:CH20 and PM6:CH21, PM6:CH22 blended films have higher and more balanced electron/hole mobilities. How about the mobilities in pristine films of three NFAs? Additionally, the device condition should be described more detailly, such as rotation speeds and so on.
3. The increased dielectric constants of brominated CH22 are quite important to improve its performance of resulting OSCs. The authors should provide more detailed information about device structures used in the process of obtaining dielectric constants.
4. Since PM6:CH22 system exhibited relatively good stabilities, I suggest the authors provide longer term stability test data, such as 1000 h for storage stability.
5. The following important literatures (Angew. Chem. Int. Ed. 2022 ,61, e202209454, J. Mater. Chem. A., 2020, 8, 4856, and Sol. RRL, 2020, 4, 2000212.) related to the fluorinated or brominated ending groups are suggested to add.

Reviewer #2 (Remarks to the Author):

In this manuscript, the author reported three NFAs by brominating on central unit and claimed the introduction of Br can transform the intermolecular packing modes. Systemic investigations were conducted to prove the effect of bromination on polarizability, dipole moment and relative dielectric constant etc. Through XRD measurements and DFT calculation, the molecular geometries and

intermolecular packing modes were carefully analyzed and calculated. Interestingly, from CH20 to CH22, the Br on central unit can transform the intermolecular packing mode from “C/C” to effective “E/E” mode. The authors fabricated OSC devices featured with a conventional architecture and studied their photovoltaic performance in detail. As a result, devices based on PM6:CH22 showed a highest PCE up to 19.06% with improved thermal stability and thick-film performance (500 nm, 15.7%). Overall, the author subtly adopted the strategy of bromination on central unit of A-D-A type acceptors to increase the variety and achieve more excellent performance. I believe that this work will arouse much interest of the community, and it is also well organized with decent analyses and reveals some interesting results. Thus, I would like to recommend it to be published in Nature Communications after some revisions and answer below questions.

1) In page 4, the description “In addition, the possibly halogen bonds ($X\cdots H$, $X\cdots S$, $X\cdots \pi$, etc.) induced by F or Cl involved secondary ...” was less rigorous. $X\cdots H$ can stand for halogen bonds, but $X\cdots S$ and $X\cdots \pi$ can be described as non-covalent interactions. In page 4, the description “The large and loose outmost electron cloud makes bromine easily polarizing...” was a false statement. It should be corrected as “The large and loose outmost electron cloud makes bromine easily polarized”. In page 5, the description “...easily synthesizing and polarizing” should be corrected as “... easily synthesized and polarized”. In the page 20, “exciton dissociation efficiency (P_{diss}) and charge collection efficiency (P_{coll}) were...” should be consistent with Figure 5(c). In page 10, the description “blue-shifted λ_{max} as mentioned above (Figure 1d)”, should be corrected as “blue-shifted λ_{onset} as mentioned above (Figure 1d)”. In the page 15, the description “...performed to delicately tuning both molecular structures and interocular packing modes” should be corrected as “...performed to delicately tuning both molecular structures and intermolecular packing modes”. Please correct it and carefully check these details.

2) In page 4, the description “Thirdly, brominated compounds are easy to synthesize under a mild condition and relatively low cost comparing to fluoride and chloride, at the same time, stable enough when applied in light-harvesting materials”. Generally speaking, bromides are often used as important intermediates in the synthesis of other organic compounds. And the bond length of C-Br is longer than other bond length of C-X (X = F, Cl). Please comment on whether it decrease the photostability of these acceptors?

3) In page 11, the description “S...N secondary interactions further guarantee the relatively planar conjugated backbones.” How did the author confirm the interaction between S and N? Please check the van der Waals Radii of S and N. In general, certain conditions need to be met to prove the interactions.

4) In the page 15, the author claimed that “As shown in Figure S10 and Table S5, CH22-6Br only achieved a PCE of 17.00% with significantly decreased JSC and FF comparing to those of CH22, which may be caused by not only the excessive molecular aggregation, but also the inefficient molecule packing of “E/E” mode due to the relatively larger steric hindrance of bromine than its homomorphic fluorine or chlorine.” However, in the text and SI, there's no data to prove the feature of excessive molecular aggregation and inefficient molecule packing of “E/E” mode. Please list the relevant evidences and make comments.

5) In this work, by brominating on central unit, the PCE of OSCs based on PM6:CH22 is up to 19.06%. In your previous work, the fluorination and chlorination on the central unit have been studied respectively.

But the PCE cannot reach up to 19%. And whether it exists that halogen atoms change the pattern of intermolecular packing modes. The Intrinsic characteristic or the device processing, which one is more important and how can you explain the discrepancy of different halogenation strategies?

Reviewer #3 (Remarks to the Author):

The authors of this manuscript synthesized new A-DA'D-A type small molecule acceptors (SMAs) CH21 and CH22 with bromine substitution on their central A' unit, and compared the physicochemical and photovoltaic properties of the brominated SMAs with the corresponding SMA CH20 without the bromination. They found that the brominated SMAs CH21 and CH22 possess larger relative dielectric constant, smaller exciton binding energy, higher electron mobility and high photovoltaic performance than CH20. Importantly, the CH22-based organic solar cells (OSCs) with PM6 as polymer donor demonstrated a high power conversion efficiency (PCE) of 19.06%. The results are very interesting for the related researchers. I think this manuscript can be accepted for publication after some minor revisions as indicated in the following:

(1) I suggest to revise the name of "non-fullerene acceptor" (NFA) to "small molecule acceptor" (SMA), because no researcher use fullerene derivatives as acceptor in the research field of OSCs at present, so that no need to emphasize the non-fullerene acceptors.

(2) Photo-stability of the SMAs is very important for future application, I suggest the authors to compare the photo-stability of the SMAs with and without the bromine substitution.

Point-by-point response to the reviewers' comments for Nature Communications

(Manuscript NCOMMS-23-13602)

(Texts in blue are our replies)

Reviewer #1 (Remarks to the Author):

Chen and co-authors constructed three A-D-A type NFAs with brominating on central unit instead of end groups. Therefore, the advantages of bromine can be maximized whilst circumventing its adverse effects of large steric hindrance on molecular packing, especially for “end unit to end unit” packing mode. By keeping a superior 3D molecular packing network, PM6:CH22-based binary OSCs eventually reach the highest efficiency of 19.06% for brominated NFAs-based devices along with a good stability. Moreover, an excellent efficiency of 15.70% was also achieved when further thickening active layers up to 500 nm. The authors indeed exhibit the great potentials for achieving high-performance OSCs through further brominating on NFAs, by developing such a rare case of brominated NFA. In addition, the manuscript has been well organized and the resulting conclusions are quite convincing at the current stage. Based on these considerations above, I recommended its publication in Nature Communications after a minor revision. Several suggestions are following:

Reply: Thank you very much for your positive comments and valuable suggestions to improve our manuscript.

1. In this text, the highest efficiency of 19.06% for brominated NFAs has been reported based on PM6:CH22. Comparisons with other brominated OSCs should be also mentioned (perhaps in an additional Table).

Reply: Thank you very much for the valuable suggestions. We have added the comparisons with other brominated OSCs in “Revised Supplementary Information, Supplementary Table 10”. You can also find it below:

Supplementary Table 10. Comparison of brominated OSCs performance between this work and references.

Active layer	V_{oc} (V)	J_{sc} (mA cm ⁻²)	FF (%)	PCE (%)	Ref.
--------------	--------------	---------------------------------	--------	---------	------

PTPDBDT:Br-ITIC	0.93	15.4	66.00	9.4	16
PBDB-T:F-Br	0.87	18.22	76.00	12.05	17
PBDB-T:IT-2Br	0.83	17.93	71.00	10.66	18
PM6:ITIC-2Br- γ	0.89	19.01	71.21	12.05	19
PM6:ITC-2Br2	0.90	19.8	73.80	13.10	20
PBDB-T:BTTPC-Br	0.86	24.71	71.00	15.22	21
PM6:TSeIC4Br	0.77	21.27	72.40	11.92	22
PM7:BDSe-2(BrCl)	0.83	22.91	76.50	14.54	23
PM6:ZB	0.90	26.38	64.16	15.23	24
PM6:BTIC-2Br-m	0.88	25.03	73.13	16.11	25
PM6:BTP-ClBr	0.906	23.48	79.00	16.82	26
PM6:C8IDT-Br	0.97	15.66	64.81	9.85	27
PM6:BTIC-4EO-4Br	0.84	22.78	65.21	12.41	28
PM6:BTP-(Br,Me)-1	0.92	21.38	68.25	13.43	29
PM6:IT-2Br	0.845	21.98	69.53	12.92	30
PTQ10:IDIC-Br	0.92	16.3	65.00	10.80	31
PBDB-T:6TIC-2Br	0.76	22.74	68.27	11.77	32
D18:F-ThBr	1.089	16.68	71.69	13.03	33
PM6:Y-BO-FBr	0.85	25.83	75.02	16.47	34
PM6:BTP-H2	0.932	25.33	78.50	18.50	35
PM6:CH22-6Br	0.871	26.25	74.31	17.00	This work
PM6:CH21	0.873	26.57	78.13	18.12	This work
PM6:CH22	0.884	26.74	80.62	19.06	This work

16. Yang F, Li C, Lai W, Zhang A, Huang H, Li W. Halogenated conjugated molecules for ambipolar field-effect transistors and non-fullerene organic solar cells. *Mater. Chem. Front.* **1**, 1389–1395 (2017).
17. Wang Y, *et al.* A halogenation strategy for over 12% efficiency nonfullerene organic solar cells.

- Adv. Energy Mater.* **8**, 1702870 (2018).
18. Lu S, *et al.* Halogenation on terminal groups of itic based electron acceptors as an effective strategy for efficient polymer solar cells. *Sol. Energy* **195**, 429–435 (2020).
 19. Qu J, *et al.* Bromination of the small-molecule acceptor with fixed position for high-performance solar cells. *Chem. Mater.* **31**, 8044–8051 (2019).
 20. Luo Z, *et al.* Significantly improving the performance of polymer solar cells by the isomeric ending-group based small molecular acceptors: Insight into the isomerization. *Nano Energy* **66**, 104146 (2019).
 21. Qin R, *et al.* Tuning terminal aromatics of electron acceptors to achieve high-efficiency organic solar cells. *J. Mater. Chem. A*, **7**, 27632–27639 (2019).
 22. Zhang C, *et al.* Tetrabromination versus tetrachlorination: A molecular terminal engineering of nonfluorinated acceptors to control aggregation for highly efficient polymer solar cells with increased voc and higher jsc simultaneously. *Sol. RRL* **4**, 2000212 (2020).
 23. Wan S-S, *et al.* A bromine and chlorine concurrently functionalized end group for benzo[1,2-b:4,5-b']diselenophene-based non-fluorinated acceptors: A new hybrid strategy to balance the crystallinity and miscibility of blend films for enabling highly efficient polymer solar cells. *J. Mater. Chem. A*, **8**, 4856-4867 (2020).
 24. Zhang M, *et al.* Effects of monohalogenated terminal units of non-fullerene acceptors on molecular aggregation and photovoltaic performance. *Sol. Energy* **208**, 866–872 (2020).
 25. Wang H, *et al.* Bromination: An alternative strategy for non-fullerene small molecule acceptors. *Adv. Sci.* **7**, 1903784 (2020).
 26. Luo Z, *et al.* Altering the positions of chlorine and bromine substitution on the end group enables high-performance acceptor and efficient organic solar cells. *Adv. Energy Mater.* **10**, 2002649 (2020).
 27. Zhang L, Tu S, Wang W, Ling Q. Brominated small-molecule acceptors with a simple non-fused framework for efficient organic solar cells. *ACS Appl. Energ. Mater.* **4**, 4805–4814 (2021).
 28. Mo D, Chen H, Zhu Y, Huang H-H, Chao P, He F. Effects of halogenated end groups on the performance of nonfullerene acceptors. *Acs Appl. Mater. Interfaces* **13**, 6147–6155 (2021).
 29. Chen L, *et al.* End-group modifications with bromine and methyl in nonfullerene acceptors: The effect of isomerism. *Acs Appl. Mater. Interfaces* **13**, 29737–29745 (2021).
 30. Liu Y, *et al.* Effects of brominated terminal groups on the performance of fused-ring electron acceptors in organic solar cells. *Dyes. Pigments* **194**, 109652 (2021).
 31. Radford CL, Mudiyansele PD, Stevens AL, Kelly TL. Heteroatoms as rotational blocking groups for non-fullerene acceptors in indoor organic solar cells. *ACS Energy Lett.* **7**, 1635–1641 (2022).
 32. Liu X, *et al.* Near-infrared nonfullerene acceptors with halogenated terminated fused tris(thienothiophene) for efficient polymer solar cells. *Sol. Energy* **231**, 433–439 (2022).
 33. Huang Y, *et al.* Tandem organic solar cells with 18.67% efficiency via careful subcell design and selection. *J. Mater. Chem. A*, **10**, 11238–11245 (2022).
 34. Wang L, *et al.* Non-fullerene acceptors with hetero-dihalogenated terminals induce significant difference in single crystallography and enable binary organic solar cells with 17.5% efficiency. *Energy Environ. Sci.* **15**, 320–333 (2022).
 35. He C, *et al.* Manipulating the D:A interfacial energetics and intermolecular packing for 19.2% efficiency organic photovoltaics. *Energy Environ. Sci.* **15**, 2537–2544 (2022).

2. Compared with PM6:CH20 and PM6:CH21, PM6:CH22 blended films have higher and more balanced electron/hole mobilities. How about the mobilities in pristine films of three NFAs? Additionally, the device condition should be described more details, such as rotation speeds and so on.

Reply: Thank you very much for your suggestions.

As advised, we have added the measurement of the electron mobilities for pristine films of three SMAs by using the space-charge limited current (SCLC) method in the revised supporting information. And the corresponding experimental electron mobilities of CH20, CH21 and CH22 are 2.14×10^{-4} , 3.45×10^{-4} and $4.63 \times 10^{-4} \text{ cm}^2 \text{ V}^{-1} \text{ s}^{-1}$, respectively (see Supplementary Fig. 13a-b below). The largest mobility of CH22 among three SMAs agrees well with largest relative dielectric constant, and also is beneficial for the efficient charge transport. We have also discussed the mobilities in pristine films of three SMAs in “Revised Manuscript” as following: “Note that the favorable molecular packings endow with CH22-based neat film an improved electron mobility (μ_e , Supplementary Fig. 13a-b) of $4.63 \times 10^{-4} \text{ cm}^2 \text{ V}^{-1} \text{ s}^{-1}$, comparing to that of CH20 ($2.14 \times 10^{-4} \text{ cm}^2 \text{ V}^{-1} \text{ s}^{-1}$) and CH21 ($3.45 \times 10^{-4} \text{ cm}^2 \text{ V}^{-1} \text{ s}^{-1}$).”

Supplementary Fig. 13 Mobilities of the neat and blended films of three SMAs. (a) Electron mobilities of SMAs. (b) Electron mobilities distributions counted by 5 devices of SMAs. (c) Electron mobilities and (d) hole mobilities of blended films.

Also, the detailed device fabrication conditions have been added in “Revised Supporting Information”. You can also find them as following: “**Space-Charge-Limited Current (SCLC) Measurement.** The SCLC method was used to measure the hole and electron mobilities, by using a diode configuration of ITO/PEDOT:PSS/active layer/MoO₃/Ag for hole and ITO/ZnO/active layer/PNDIT-F3N/Ag for electron. The SMAs were fully dissolved in CF with 15 mg mL⁻¹ and then the solutions were stirred 4 hours at room temperature and spin-casted at 1500 rpm for 30s. After spin-coating, the neat films were annealed at 80 °C for 5 mins. The fabrication method of blended films was consistent with that of active layer of device.”

3. The increased dielectric constants of brominated CH22 are quite important to improve its performance of resulting OSCs. The authors should provide more detailed information about device structures used in the process of obtaining dielectric constants.

Reply: Thank you very much for your comments. The details information about device structures for the dielectric constant measurements have been added in “Revised Supporting Information”, you can also find them as following: “**Relative Dielectric Constant (ϵ_r) Test.** The dielectric constant should be calculated in terms of the material’s geometric capacitance, which represents the capacitance derived from only the material itself (the electronic, atomic, and ionic polarization). The capacitance-frequency of CH20, CH21 and CH22 neat and blended films were performed with a capacitor architecture of ITO/active layer/Ag at difference frequency from 100 Hz to 1M Hz using the Zennium-E under dark conditions and analyzed with the Zahner Analysis software.”

4. Since PM6:CH22 system exhibited relatively good stabilities, I suggest the authors provide longer term stability test data, such as 1000 h for storage stability.

Reply: Thank you very much for your valuable suggestions. We further evaluated the storage stability of PM6:CH22-based OSCs for over 1500 h, and found that above 96% of its initial PCEs could be maintained. Simultaneously, we also evaluated the longer thermal stability with 400 h and found that ~85% of its initial PCEs could be

maintained. The corresponding data have been added in “Revised Manuscript” and “Revised Supporting Information, Supplementary Fig. 15”, respectively. You can also find them as following: “It is also worth noting that PCEs of PM6:CH22-based OSCs could be maintained above 96% and ~85% compared to its initial PCEs after 1500 h under room temperature and 400 h under heat treatment at 65 °C (Supplementary Fig. 15), respectively.”

Supplementary Fig. 15 Stability. PCE variation versus operating time in a glovebox filled with nitrogen at (a) room temperature and (b) 65 °C of PM6:CH22-based devices.

5. The following important literatures (Angew. Chem. Int. Ed. 2022 ,61, e202209454, J. Mater. Chem. A., 2020, 8, 4856, and Sol. RRL, 2020, 4, 2000212.) related to the fluorinated or brominated ending groups are suggested to add.

Reply: Thank you so much for your efforts to improve our paper. We have added the references related to the fluorinated or brominated ending groups (ref. 40 and ref. 45-46) in “Revised Manuscript”. The newly added references are also showed below:

40. Yan L, *et al.* Regioisomer-free difluoro-monochloro terminal-based hexa-halogenated acceptor with optimized crystal packing for efficient binary organic solar cells. *Angew. Chem. Int. Ed.* **61**, e202209454 (2022).
45. Wan S-S, *et al.* A bromine and chlorine concurrently functionalized end group for benzo[1,2-b:4,5-b']diselenophene-based non-fluorinated acceptors: A new hybrid strategy to balance the crystallinity and miscibility of blend films for enabling highly efficient polymer solar cells. *J. Mater. Chem. A.* **8**, 4856-4867 (2020).
46. Zhang C, *et al.* Tetrabromination versus tetrachlorination: A molecular terminal engineering of nonfluorinated acceptors to control aggregation for highly efficient polymer solar cells with increased voc and higher jsc simultaneously. *Sol. RRL* **4**, 2000212 (2020).

Reviewer #2 (Remarks to the Author):

In this manuscript, the author reported three NFAs by brominating on central unit and claimed the introduction of Br can transform the intermolecular packing modes. Systemic investigations were conducted to prove the effect of bromination on polarizability, dipole moment and relative dielectric constant etc. Through XRD measurements and DFT calculation, the molecular geometries and intermolecular packing modes were carefully analyzed and calculated. Interestingly, from CH20 to CH22, the Br on central unit can transform the intermolecular packing mode from “C/C” to effective “E/E” mode. The authors fabricated OSC devices featured with a conventional architecture and studied their photovoltaic performance in detail. As a result, devices based on PM6:CH22 showed a highest PCE up to 19.06% with improved thermal stability and thick-film performance (500 nm, 15.7%). Overall, the author subtly adopted the strategy of bromination on central unit of A-D-A type acceptors to increase the variety and achieve more excellent performance. I believe that this work will arouse much interest of the community, and it is also well organized with decent analyses and reveals some interesting results. Thus, I would like to recommend it to be published in Nature Communications after some revisions and answer below questions.

Reply: Thank you very much for your positive comments and efforts to improve our manuscript.

1) In page 4, the description “In addition, the possibly halogen bonds ($X\cdots H$, $X\cdots S$, $X\cdots\pi$, etc.) induced by F or Cl involved secondary ...” was less rigorous. $X\cdots H$ can stand for halogen bonds, but $X\cdots S$ and $X\cdots\pi$ can be described as non-covalent interactions.

In page 4, the description “The large and loose outmost electron cloud makes bromine easily polarizing...” was a false statement. It should be corrected as “The large and loose outmost electron cloud makes bromine easily polarized”.

In page 5, the description “...easily synthesizing and polarizing” should be corrected as “... easily synthesized and polarized”.

In the page 20, “exciton dissociation efficiency (P_{diss}) and charge collection efficiency (P_{coll}) were...” should be consistent with Figure 5(c).

In page 10, the description “blue-shifted λ_{\max} as mentioned above (Figure 1d)”, should be corrected as “blue-shifted λ_{onset} as mentioned above (Figure 1d)”.

In the page 15, the description “...performed to delicately tuning both molecular structures and interocular packing modes” should be corrected as “...performed to delicately tuning both molecular structures and intermolecular packing modes”. Please correct it and carefully check these details.

Reply: Thank you so much for the very detailed and great suggestions! We have carefully checked the full manuscript and modified all the inappropriate description including the issues mentioned above.

2) In page 4, the description “Thirdly, brominated compounds are easy to synthesize under a mild condition and relatively low cost comparing to fluoride and chloride, at the same time, stable enough when applied in light-harvesting materials”. Generally speaking, bromides are often used as important intermediates in the synthesis of other organic compounds. And the bond length of C-Br is longer than other bond length of C-X (X = F, Cl). Please comment on whether it decrease the photostability of these acceptors?

Reply: Thank you very much for your valuable suggestions. Indeed, the bond length of C-Br (~1.88 Å) is longer than other bond lengths of C-X (X=F, ~1.34 Å; Cl, ~1.73 Å). To demonstrate the photo stability of CH22 with brominating, we have recorded the UV-vis spectra after continuous illumination under one sun for non-brominated CH20 and stepwise brominated CH21 and CH22. Simultaneously, we also confirmed the photo stability of brominated CH22 by nuclear magnetic resonance (NMR) spectroscopy after continuous illumination.

Supplementary Fig. 9 displays the light spectrum of LEDs used for stability test and Supplementary Fig. 10 represents the UV-vis absorption spectra of three acceptors in both solution and solid films. After continuous illumination of 193 h in toluene and 581 h in films, all the three acceptors demonstrate negligible changes of UV-vis absorption spectra, demonstrating that the C-Br bond does not decrease the photostability of acceptors. Please note that the only ~3 nm shift of the maximum absorption peaks for all the three acceptors in solid films may be attributed to the changes of nanoscale morphology during the continuous illumination. Moreover, the

photo stability of CH22 was further confirmed by nuclear magnetic resonance (NMR) spectroscopy (Supplementary Fig. 11). There is almost no change before and after aging for over 450 h under one sun illumination, suggests that CH22 is stable enough under light exposure.

The detailed about photo stability for the three SMAs have been added in “Revised Manuscript” and “Revised Supporting Information, Supplementary Fig. 9-11” respectively. You can also find them as following: “Moreover, all the three SMAs also exhibit excellent photo stability indicated by their UV-vis spectra. As shown in Supplementary Fig. 10, the shape and intensity of absorption spectra display almost no changes after aging for over 550 h under one sun illumination.⁶³ Simultaneously, the photo stability of CH22 was further confirmed by nuclear magnetic resonance (NMR) spectroscopy. There is no change before and after aging for over 450 h under one sun illumination (Supplementary Fig. 11), which suggests that the introduction of C-Br bonds does not decrease the photo stability of acceptors.⁶⁴ The excellence thermal stability and photo stability of these three SMAs can meet well the chemical stability requirement as light absorption materials.”

63. Liu T, *et al.* Photochemical decomposition of γ -series non-fullerene acceptors is responsible for degradation of high-efficiency organic solar cells. *Adv. Energy Mater.* **13**, 2300046 (2023).
64. Li Y, *et al.* Non-fullerene acceptor organic photovoltaics with intrinsic operational lifetimes over 30 years. *Nat. Commun.* **12**, 5419 (2021).

Supplementary Fig. 9 Light spectrum of the LEDs used for stability test in this work as extracted from the company’s datasheet.

Supplementary Fig. 10 Photo stability. UV-vis absorption spectra plotted vs. aging time under continuous illumination of (a-c) CH20, CH21 and CH22 in dilute toluene solutions and (d-f) CH20, CH21 and CH22 in films.

Supplementary Fig. 11 Photo stability. ¹H NMR spectra of CH22 fresh and aged samples in CDCl₃. The samples of spin-coated films of CH22 were aged under continuous illumination for 468 h, then collected and dried before conducting ¹H NMR.

3) In page 11, the description “S⋯N secondary interactions further guarantee the relatively planar conjugated backbones.” How did the author confirm the interaction between S and N? Please check the van der Waals Radii of S and N. In general, certain conditions need to be met to prove the interactions.

Reply: Thank you very much for the valuable suggestions. We strongly agree with the reviewer that the interaction between S and N should meet certain conditions.

Firstly, to confirm the potential interactions, the van der Waals Radii of S and N were checked and the sum van der Waals radii of S and N is ~ 3.55 Å when using the values of Alvarez. Note that the distances of S⋯N are ~ 3.3 - 3.4 Å for all the three acceptors based on our single crystal analysis, which are shorter than the sum van der Waals radii of S and N, indicating the non-bonding interactions between S and N in these three SMAs.

Secondly, the noncovalent intramolecular interactions of S⋯N can be also confirmed by the reduced density gradient (RDG) analysis (see Supplementary Fig. 12 below). It can be observed that the light blue iso-surfaces appear between the S and N atoms for CH20, CH21 and CH22, indicating the existence of the S⋯N attraction interactions.

The details about the existing intramolecular noncovalent S⋯N interactions for these three SMAs have been added in “Revised Manuscript” and “Revised Supporting Information, Supplementary Fig. 12”, respectively. You can also find them as following: “Note that the distances of S⋯N between phenazine and bridged thiophene are ~ 3.3 - 3.4 Å for all the three acceptors, slightly smaller than the sum van der Waals radii (~ 3.55 Å when using the values of Alvarez⁶⁵) of S and N, indicating the possible noncovalent interactions between S and N in these SMAs. This could also be supported by the reduced density gradient (RDG) analysis indicated by the light blue iso-surfaces between S and N atoms (Supplementary Fig. 12).^{66, 67, 68}”

65. Politzer P, Murray JS. The use and misuse of van der waals radii. *Struct. Chem.* **32**, 623–629 (2021).

66. Johnson ER, Keinan S, Mori-Sánchez P, Contreras-García J, Cohen AJ, Yang W. Revealing noncovalent interactions. *J. Am. Chem. Soc.* **132**, 6498–6506 (2010).

67. Lu T, Chen F. Multiwfn: A multifunctional wavefunction analyzer. *J. Comput. Chem.* **33**, 580–592 (2012).

68. Yang K, *et al.* Intramolecular noncovalent interaction-enabled dopant-free hole-transporting materials for high-performance inverted perovskite solar cells. *Angew. Chem. Int. Ed.* **61**, e202113749 (2022).

Supplementary Fig. 12 Reduced density gradient (RDG) versus $\text{sign}(\lambda_2)\rho$ with a (RDG) iso-surface of the central phenazine cores and bridged thiophene structures of (a, b) CH20, (c, d) CH21 and (e, f) CH22.

4) In the page 15, the author claimed that “As shown in Figure S10 and Table S5, CH22-6Br only achieved a PCE of 17.00% with significantly decreased J_{SC} and FF comparing to those of CH22, which may be caused by not only the excessive molecular aggregation, but also the inefficient molecule packing of “E/E” mode due to the relatively larger steric hindrance of bromine than its homomorphic fluorine or chlorine.” However, in the text and SI, there's no data to prove the feature of excessive molecular aggregation and inefficient molecule packing of “E/E” mode. Please list the relevant evidences and make comments.

Reply: Thank you very much for your valuable suggestions to improve our manuscript.

As regards to the excessive molecular aggregations, we have studied the nanoscale molecular aggregated morphology and packing orientations of CH22-6Br using atomic force microscopy (AFM) and grazing incidence wide angle X-ray scattering (GIWAXS).

As in the newly added Supplementary Fig. 17 and also shown below, the fiber diameter size of 12.7 nm can be observed based on a statistical analysis of PM6:CH22-6Br, which is larger than that of 11.9 nm for PM6:CH22 blend. Moreover, a larger crystal coherence length (CCL) of 23.37 Å of CH22-6Br neat film can be estimated from GIWAXS (See Supplementary Fig. 18 and Supplementary Table 6 below) comparing to that of CH22 (22.53 Å). To sum up, the large fiber size in PM6:CH22-6Br blend film and CCL in CH22-6Br neat film could support the excessive molecular aggregation of CH22-6Br with respect to that of CH22.

Supplementary Fig. 17 AFM characterization. (a) AFM height images and phase images of PM6:CH22 and PM6:CH22-6Br blended films. (b) The statistical distribution of the fibril diameter for PM6:CH22 and PM6:CH22-6Br blended films. (c, d) The line profile to obtain the fibril width from the AFM phase images of PM6:CH22 and PM6:CH22-6Br blended films.

As regards to the inefficient molecule packing of “E/E” mode, we measured the GIWAXS of CH22-6Br in both neat and blended films (See Supplementary Fig. 18 and Supplementary Table 6 below). The π - π stacking distances can be observed for CH22-6Br (3.74 and 3.70 Å for neat and blended films, respectively), which are both slightly larger than that of 3.66 and 3.63 Å for CH22 neat and blended films. Given that the only structural difference of CH22 and CH22-6Br locates on the brominating on end units, the less compact π - π stacking should be caused by the end units in theory, most probably the inefficient molecular packing of “E/E” mode. In order to get a more rigorous statement, we have modified the description of “inefficient molecular packing of “E/E” mode” with “the less compact π - π stacking of molecules”.

Supplementary Fig. 18 GIWAXS characterization. (a, b) 2D GIWAXS patterns of CH22-based neat and blended films. (c) The corresponding out of plane (OOP) and in plane (IP) extracted line-cut profiles of CH22-based neat and blended films. (d, e) 2D GIWAXS patterns of CH22-6Br-based neat and blended films. (f) The corresponding out of plane (OOP) and in plane (IP) extracted line-cut profiles of CH22-6Br-based neat and blended films.

Supplementary Table 6. The detailed parameters of corresponding 2D GIWAXS.

Materials	(010) Diffraction Peak				(100) Diffraction Peak	
	q (\AA^{-1})	d ^[a] (\AA)	FWHM (\AA^{-1})	CCL ^[b] (\AA)	q (\AA^{-1})	d ^[a] (\AA)

CH22	1.718	3.66	0.251	22.53	0.410	15.33
PM6:CH22	1.733	3.63	0.252	22.44	0.408	15.42
CH22-6Br	1.678	3.74	0.242	23.37	0.405	15.51
PM6:CH22-6Br	1.700	3.70	0.236	23.96	0.400	15.71

^aCalculated from the equation: $d\text{-spacing}=2\pi/q$. ^bObtained from the Scherrer equation: $CCL=2\pi K/FWHM$, where FWHM is the full-width at half-maximum and K is a shape factor (K= 0.9 here).

The detailed measurement information and discussions about CH22-6Br have been added in “Revised Manuscript” and “Revised Supporting Information, Supplementary Fig. 17-18 and Supplementary Table 6”, respectively. You can also find them as following: “The excessive molecular aggregation of CH22-6Br can be confirmed by the relatively larger fiber size (12.7 nm for PM6:CH22-6Br, 11.9 nm for PM6:CH22) based on a statistical analysis of atomic force microscopy (AFM) phase images (Supplementary Fig. 17) and slightly greater crystal coherence length (CCL) of 23.37 Å in CH22-6Br neat film estimated from GIWAXS (Supplementary Fig. 18 and Supplementary Table 6) comparing to that of CH22 (22.53 Å). Moreover, the slightly larger π - π stacking distances for CH22-6Br (3.74 and 3.70 Å for neat and blended films) can be estimated from GIWAXS (Supplementary Fig. 18 and Supplementary Table 6) comparing to that of CH22 (3.66 and 3.63 Å for neat and blended films), which may be caused by the relatively larger steric hindrance of bromine.” We have also modified our statement of “inefficient molecular packing of “E/E” mode” with “the less compact π - π stacking of molecules” in “Revised Manuscript” to get a more rigorous discussion.

5) In this work, by brominating on central unit, the PCE of OSCs based on PM6:CH22 is up to 19.06%. In your previous work, the fluorination and chlorination on the central unit have been studied respectively. But the PCE cannot reach up to 19%. And whether it exists that halogen atoms change the pattern of intermolecular packing modes. The Intrinsic characteristic or the device processing, which one is more important and how can you explain the discrepancy of different halogenation strategies?

Reply: Thank you very much for your comments and attentions to our previous work about the central unit halogenation strategy of small molecule acceptors (SMAs). Regarding to which of the intrinsic characteristic or the device processing is more important, we believe the performance difference is mainly due to the intrinsic characteristic of these molecules, and the reasons are summarized below:

(1) We totally agree with the reviewer's concern that the device processing conditions may cause the performance variations of OSCs. However, for OSCs based on SMAs with different halogenations on central units, the best photovoltaic parameters were achieved under their individual best device processing conditions (Table R1). More importantly, the statistical analysis has also been carried out, which could avoid the interference of device processing conditions to some extent (Fig. R1).

Table R1 Summary of device parameters for OSCs of PM6:CH13, PM6:CH23 and PM6:CH22 OSCs

Active layers	V_{oc} (V)	J_{sc} (mA cm ⁻²)	Cal. J_{sc} (mA cm ⁻²)	FF (%)	PCE (%)	Ref.
PM6:CH13	0.872 (0.874±0.002)	25.31 (25.13±0.34)	26.17	75.99 (73.52±0.68)	16.77 (16.58±0.14)	Energy Environ. Sci. 2022 , 15 , 3519–3533
PM6:CH23	0.876 (0.877±0.004)	26.64 (26.54±0.11)	26.17	80.45 (80.22±0.16)	18.77 (18.67±0.09)	Adv. Funct. Mater. 2023 , 2301573
PM6:CH22	0.884 (0.884±0.002)	26.74 (26.62±0.13)	26.17	80.62 (80.34±0.24)	19.06 (18.91±0.08)	This work

Fig. R1 The histogram of PCEs based on 15 devices of PM6:CH13, PM6:CH23 and PM6:CH22 fitted with Gaussian distributions (solid lines).

(2) The halogenation induced molecular packing varying could affect the charge transport dynamics, thus contributing to diverse J_{SC} and FF (Sci. China Chem. **2022**, *65*, 1362–1373; Energy Environ. Sci. **2022**, *15*, 3519–3533). Therefore, the electron mobilities (μ_e) of CH13, CH23 and CH22 neat films were measured by using the space-charge limited current (SCLC) method under the same device structure and conditions. As shown in Fig. R2 and Table R2, the corresponding average electron mobilities obtained from 5 devices of CH13, CH23 and CH22 are 2.54×10^{-4} , 4.21×10^{-4} and $4.63 \times 10^{-4} \text{ cm}^2 \text{ V}^{-1} \text{ s}^{-1}$, respectively. The brominated CH22 exhibits a relatively larger mobility than that of fluorinated CH13 and chlorinated CH23, which may could account for its best performance to some extent.

Fig. R2 Mobilities of the CH13, CH23 and CH22 SMAs. (a) Electron mobilities of SMAs. (b) Electron mobilities distributions counted by 5 devices of SMAs.

(3) In our previous work (Energy Environ. Sci. **2022**, *15*, 3519–3533), we have fully demonstrated that varying halogenations on central units will give rise to significantly effects on molecular packing behaviors, nanoscale film morphologies and even photovoltaic performances of OSCs. In order to unveil the effects of different central unit halogenations on molecular packings, we have summarized and analyzed the single-crystal structures of CH13 (fluorinated central unit; Energy Environ. Sci. **2022**, *15*, 3519–3533), CH23 (chlorinated central unit; Adv. Funct. Mater. **2023**, 2301573) and CH22 (brominated central unit) in Fig. R3 and Table R2 below. Note that CH13, CH23 and CH22 possess the same molecular backbones, however, the only structural difference is the different halogenations on their central units (Fig. 3a).

As shown in Fig. R3b, CH23 and CH22 exhibit a slightly distorted skeleton with a larger dihedral angle of $\sim 11^\circ$ between two end groups than that of $\sim 3^\circ$ for CH13. In addition, from the overall view in Fig. R3c, the single crystal of CH13 exhibits a monoclinic system with a rectangle-shaped void of $\sim 26.3 \times 14.7 \text{ \AA}$. CH23 and CH22 single crystals, which is transformed into triclinic systems, possess the rectangle-shaped voids of $\sim 18.6 \times 12.1 \text{ \AA}$ and $\sim 16.7 \times 14.0 \text{ \AA}$, respectively, smaller than that of CH13. More importantly, as displayed in Fig. R3d, 3e and 3f, CH13 has four molecular packing modes of “E/E+C/C”, “E/E-1”, “E/E-2” and “dual E/b”. However, CH23 and CH22 are quite different from CH13, being “E/C”, “E/E” and “dual C/b” mode. Note that the various packing modes have been proven to possess different intermolecular $\pi \cdots \pi$ stacking distances and electron transfer integrals (V_E), thus resulting in much different photovoltaic parameters (Energy Environ. Sci. **2022**, *15*, 3519–3533). Therefore, the PCEs of OSCs based on CH23 and CH22 have a remarkable improvement with respect to that of CH13 (Table R1), which agrees well with their quite different molecular packing behaviors. Meanwhile, it is also reasonable that OSCs based on CH22 just display a minor PCE improvement comparing to that of CH22, when taking their similar molecular packing modes into considerations.

Fig. R3 Single-crystal structures. (a) Molecular structures of CH13, CH23 and CH22. (b) Monomolecular single crystallographic structures of CH13, CH23 and CH22 in top-view and side-view (the alkyl chains are omitted for clarity). (c) Single-crystal packing diagrams from the top view of CH13, CH23 and CH22, respectively. (d, e, f) Interlayer π - π stacking distances including all the corresponding intermolecular packing modes and different intermolecular packing modes of CH13, CH23 and CH22, respectively. Red: end groups (E); grey: bridge unit (b); blue: central unit (C).

Table R2 Crystallographic and π - π interaction parameters of CH13, CH23 and CH22

Compound	Void sizes (shape)	Packing modes	$d_{\pi-\pi}$ ^[a] (Å)	Intermolecular potentials (kJ mol ⁻¹)	V_E ^[b] (meV)	$\mu_{e,avg}$ ^[c] (10 ⁻⁴ cm ² V ⁻¹ s ⁻¹)
CH13	26.3×14.7 Å	Mode 1 (E/E+C/C)	3.353/3.804	236.4	95.38	2.54
		Mode 2-1 (E/E-1)	3.477	77.8	60.89	
		Mode 2-2 (E/E-2)	3.573	85.5	7.04	
		Mode 3 (Dual E/b)	3.383	169.7	15.99	
CH23	18.6×12.1 Å	Mode 1 (E/C)	3.474	75.2	8.34	4.21
		Mode 2 (E/E)	3.346	130.1	38.75	
		Mode 3 (Dual C/b)	3.429	194.3	5.27	
CH22	16.7×14.0 Å	Mode 1 (E/C)	3.310	79.7	6.48	4.63
		Mode 2 (E/E)	3.351	129.9	54.69	
		Mode 3 (Dual C/b)	3.362	195.5	16.13	

^a $d_{\pi-\pi}$ is the π - π interlayer distance including the main types of intermolecular packing modes of CH13, CH23 and CH22, which is consistent with that of Figure 3. ^b V_E is the electron transfer integrals of the corresponding packing modes. ^c μ_e is the electron mobility of the corresponding neat films, which was measured by the SCLC method. The average μ_e calculated from 5 independent devices.

To sum up, based on the discussions above, the intrinsic characteristic of CH22 (or the discrepancy of different halogenation strategies) should be more important to the improvement of PCE in CH22-based OSCs.

Reviewer #3 (Remarks to the Author):

The authors of this manuscript synthesized new A-DA'D-A type small molecule acceptors (SMAs) CH21 and CH22 with bromine substitution on their central A' unit, and compared the physicochemical and photovoltaic properties of the brominated SMAs with the corresponding SMA CH20 without the bromination. They found that the brominated SMAs CH21 and CH22 possess larger relative dielectric constant, smaller exciton binding energy, higher electron mobility and high photovoltaic performance than CH20. Importantly, the CH22-based organic solar cells (OSCs) with PM6 as polymer donor demonstrated a high power conversion efficiency (PCE) of 19.06%. The results are very interesting for the related researchers. I think this manuscript can be accepted for publication after some minor revisions as indicated in the following:

Reply: Thank you very much for your positive comments and valuable suggestions to improve our manuscript.

(1) I suggest to revise the name of “non-fullerene acceptor” (NFA) to “small molecule acceptor” (SMA), because no researcher use fullerene derivatives as acceptor in the research field of OSCs at present, so that no need to emphasize the non-fullerene acceptors.

Reply: Thank you so much for your valuable suggestions. We have revised the name of “non-fullerene acceptor” (NFA) to “small molecule acceptor” (SMA) in “Revised Manuscript” and “Revised Supporting Information”. We also revised the title of this article as “A Rare Case of Brominated Small Molecule Acceptors for High-Efficiency Organic Solar Cell”.

(2) Photo-stability of the SMAs is very important for future application, I suggest the authors to compare the photo-stability of the SMAs with and without the bromine substitution.

Reply: Thank you very much for your valuable suggestions. To demonstrate the photo stability of CH22 with brominating, we have recorded the UV-vis spectra after continuous illumination under one sun for non-brominated CH20 and stepwise

brominated CH21 and CH22. Simultaneously, we also confirmed the photo stability of brominated CH22 by nuclear magnetic resonance (NMR) spectroscopy after continuous illumination.

Supplementary Fig. 9 displays the light spectrum of LEDs used for stability test and Supplementary Fig. 10 represents the UV-vis absorption spectra of three acceptors in both solution and solid films. After continuous illumination of 193 h in toluene and 581 h in films, all the three acceptors demonstrate negligible changes of UV-vis absorption spectra, demonstrating that the C-Br bond does not decrease the photostability of acceptors. Please note that the only ~3 nm shift of the maximum absorption peaks for all the three acceptors in solid films may be attributed to the changes of nanoscale morphology during the continuous illumination. Moreover, the photo stability of CH22 was further confirmed by nuclear magnetic resonance (NMR) spectroscopy (Supplementary Fig. 11). There is almost no change before and after aging for over 450 h under one sun illumination, suggests that CH22 is stable enough under light exposure.

The detailed about photo stability for the three SMAs have been added in “Revised Manuscript” and “Revised Supporting Information, Supplementary Fig. 9-11” respectively. You can also find them as following: “Moreover, all the three SMAs also exhibit excellent photo stability indicated by their UV-vis spectra. As shown in Supplementary Fig. 10, the shape and intensity of absorption spectra display almost no changes after aging for over 550 h under one sun illumination.⁶³ Simultaneously, the photo stability of CH22 was further confirmed by nuclear magnetic resonance (NMR) spectroscopy. There is no change before and after aging for over 450 h under one sun illumination (Supplementary Fig. 11), which suggests that the introduction of C-Br bonds does not decrease the photo stability of acceptors.⁶⁴ The excellence thermal stability and photo stability of these three SMAs can meet well the chemical stability requirement as light absorption materials.”

63. Liu T, *et al.* Photochemical decomposition of γ -series non-fullerene acceptors is responsible for degradation of high-efficiency organic solar cells. *Adv. Energy Mater.* **13**, 2300046 (2023).

64. Li Y, *et al.* Non-fullerene acceptor organic photovoltaics with intrinsic operational lifetimes over 30 years. *Nat. Commun.* **12**, 5419 (2021).

Supplementary Fig. 9 Light spectrum of the LEDs used for stability test in this work as extracted from the company's datasheet.

Supplementary Fig. 10 Photo stability. UV-vis absorption spectra plotted vs. aging time under continuous illumination of (a-c) CH20, CH21 and CH22 in dilute toluene solutions and (d-f) CH20, CH21 and CH22 in films.

Supplementary Fig. 11 Photo stability. ^1H NMR spectra of CH22 fresh and aged samples in CDCl_3 . The samples of spin-coated films of CH22 were aged under continuous illumination for 468 h, then collected and dried before conducting ^1H NMR.

REVIEWERS' COMMENTS

Reviewer #1 (Remarks to the Author):

This manuscript is ok for publish.

Reviewer #2 (Remarks to the Author):

The authors have made a reasonable response and careful revision. I recommend the publication of this manuscript in Nature Communications.

Reviewer #3 (Remarks to the Author):

The authors have revised their manuscript according to the reviewers' comments and revision opinions. Therefore I think the revised manuscript can be accepted for publication at its present form.